# A simple stress-based cliff-calving law

Tanja  Schlemm[1,2] and Anders  Levermann[1,2,3]

[1]Potsdam Institute for Climate Impact Research, Potsdam, Germany
[2]Institute of Physics and Astronomy, University of Potsdam, Potsdam, Germany
[3]Lamont-Doherty Earth Observatory, Columbia University, New York, USA

**Correspondence:** Anders Levermann (anders.levermann@pik-potsdam.de)

**Abstract.** Over large coastal regions in Greenland and Antarctica the ice sheet calves directly into the ocean. In contrast to ice-shelf calving, an increase in calving from grounded glaciers contributes directly to sea-level rise. Ice cliffs with a glacier freeboard larger than $\approx 100\,\mathrm{m}$ are currently not observed, but it has been shown that such ice cliffs are increasingly unstable with increasing ice thickness. This cliff calving can constitute a self-amplifying ice loss mechanism that may significantly alter
sea-level projections both of Greenland and Antarctica. Here we seek to derive a minimalist stress-based parametrization for cliff calving from grounded glaciers whose freeboards exceed the $100\,\mathrm{m}$ stability limit derived in previous studies. This will be an extension of existing calving laws for tidewater glaciers to higher ice cliffs.

     To this end we compute the stress field for a glacier with a simplified two-dimensional geometry from the two-dimensional Stokes equation. First we assume a constant yield stress to derive the failure region at the glacier front from the stress field
within the glacier. Secondly, we assume a constant response time of ice failure due to exceedance of the yield stress. With this strongly constraining but very simple set of assumption we propose a cliff-calving law where the calving rate follows a power-law dependence on the freeboard of the ice with exponents between 2 and 3 depending on the relative water depth at the calving front. The critical freeboard below which the ice front is stable decreases with increasing relative water depth of the calving front. For a dry water front it is, for example, $75\,\mathrm{m}$. The purpose of this study is not to provide a comprehensive
calving law, but to derive a particularly simple equation with a transparent and minimalist set of assumptions.

## 1   Introduction

Ice loss from Greenland and Antarctica is increasingly contributing to global sea-level rise (Rignot et al., 2014; Shepherd et al., 2018; WCRP Global Sea Level Budget Group, 2018). A possible additional future mass loss from these ice sheets is
of crucial importance for future sea-level projections (Slangen et al., 2017; Church et al., 2013; DeConto and Pollard, 2016; Kopp et al., 2017; Mengel et al., 2016; Ritz et al., 2015; Levermann et al., 2014). Ice sheet gain mass by the surface mass balance. The question whether they contribute to changes in sea level is determined by the question how strongly this mass addition is compensated or overcompensated by mass loss. Both ice sheet in Greenland and Antarctica currently show a net ice

loss. Calving accounts for roughly half the ice loss of the Antarctic ice shelves, the rest is lost by basal melt (Depoorter et al., 2013). For the Greenland ice sheet, calving accounted for two-thirds of the ice loss between 2000 and 2005, the rest is due to enhanced surface melting and runoff (Rignot and Kanagaratnam, 2006). Because surface melt increased faster than glacier speed, calving accounted for one-third of the Greenland ice sheet mass loss between 2009 and 2012 (Enderlin et al., 2014).

Tidewater glaciers calve vigorously when they are near floatation thickness producing icebergs with a horizontal extent smaller than the ice thickness. This has been expressed in semi-empirical height-above-floatation calving laws (Meier and Post, 1987; van Der Veen, 1996; Vieli et al., 2002). Calving at ice-shelf fronts or floating glacier tongues has long rest periods interrupted by the calving of large tabular ice bergs (Lazzara et al., 1999) and is preceded by the formation of deep crevasses upstream (Joughin and MacAyeal, 2005). The distinction between these two kinds of calving is not always easy because a tidal

glacier can form or lose a floating tongue; this has for example been observed at the Columbia glacier in Alaska (Walter et al., 2010).

In order to model calving not just for single glaciers but for whole ice sheets, a calving parametrization is needed. Theories describing the nucleation and spreading of crevasses in ice (Pralong and Funk, 2005) are computationally very intense and difficult to apply in simulations on long timescales and large spatial dimensions. In order to parametrize calving processes

several approaches have been used:

First, calving can be described as a function of strain rate and crevasse depth. Nye (1957) first described the formation of crevasses as a result of velocity gradients: The depth of the crevasse is determined by the strain-rate and overburdening pressure of the ice. Observations show that ice velocities are greater near the calving front than upstream (Meier and Post, 1987), hence crevasses form mainly at the calving front. When crevasses are deep enough. Icebergs are then separated from the glacier

and calve off. Benn et al. (2007) proposed a calving law with the assumption that a glacier calves where crevasses reach the water level, Nick et al. (2010) proposed calving when surface and basal crevasses meet. These calving laws have been applied successfully in 1D flow-line models (Nick et al., 2010) and in a 3D Full Stokes model (Todd et al., 2018).

Second, a number of approaches have been taken to analyze calving processes via the stress balance. Bassis and Walker (2011) analyzed depth-averaged stresses at the calving front. Considering tensile and shear failure, they found that there is an

upper limit for the thickness of stable ice cliffs: an ice cliff is only stable if the glacier's freeboard (ice thickness minus water depth) is lower than 200m. The limit decreases to 100m if weakening of the ice through crevasses is also considered. Krug et al. (2014) used damage and fracture mechanics to model calving. This approach using linear elastic fracture mechanics has recently been analyzed by Jiménez and Duddu (2018) who found that it can be applied to floating shelves but not to grounded glaciers. Morlighem et al. (2016) give a calving rate in terms of ice velocity and the von Mises stress. Recent works by Ma et al.

(2017) and Benn et al. (2017) solved the 2D full-Stokes equation at the calving front with finite element methods. Ma et al. (2017) found that while sliding glaciers calve through tensile failure, for glaciers frozen to the bed shear failure dominates. Benn et al. (2017) used finite element models to solve the stress balance and a discrete element model to simulate fracture formation. They modelled a range of calving mechanisms including calving driven by buoyancy and melt-undercutting, but did not give parametrizations of calving rates.

Finally, Mercenier et al. (2018) analyzed tensile failure with 2D finite elements and derived a calving law for tidewater glaciers. They analyzed crevasse formation at the glacier terminus, determined the distance of the crevasse to the front and the time to failure until the crevasse penetrates the whole glacier and the iceberg in front of the crevasse calves off. Together this gives an equation for the calving rate as a function of water depth and ice thickness.

All these approaches agree on the basic physics of glacier calving: Thicker ice at the terminus leads to higher stresses and larger calving rates. Glaciers terminating in water are stabilized by the water's back-pressure and have smaller calving rates.

The stability limit derived by Bassis and Walker (2011) lead to the formation of the marine ice cliff instability hypothesis. If cliff calving from ice cliffs whose freeboards exceed the stability limit is initiated in an overdeepend basin, e.g. in East Antarctica, it can lead to runaway cliff calving where higher ice cliffs are exposed the further the grounding line retreats, causing even larger cliff calving rates.

Pollard et al. (2015) and DeConto and Pollard (2016) incorporated cliff calving in Antarctica projections by assuming a linear relation between freeboard exceeding the stability limit and calving rate and showed that the marine ice cliff instability can lead to much faster sea level rise than found in previous approaches. Bassis et al. (2017) rewrote the condition that the glacier freeboard should not exceed the stability limit as a lower bound on the rate of terminus advance or equivalently an upper bound on the calving rate. More research is needed and especially a more physically based cliff calving law. Studies by Ma et al. (2017), Benn et al. (2017) and Mercenier et al. (2018) were made for tidewater glaciers not exceeding the stability limit and might not be applicable to glaciers exceeding the stability limit.

In this study, we analyze stresses at the calving front by solving the 2D Stokes equation with a finite element model in order to propose a simple cliff calving law. The purpose of this study is not to provide a comprehensive analysis. By contrast, we seek a minimalistic set of assumptions that paths the way to a simple stress-based cliff calving law.

## 2 Stress balance near the calving front

### 2.1 Problem set-up: 2D Stokes equation and boundary condition

In this study we consider a plane, flat glacier of constant thickness $H$ terminating in water of depth $D$ in a one-dimensional (flow-line) model with horizontal coordinate $x$ and vertical coordinate $z$ (Figure 1).

In order to compute the stress field near the calving front we set the glacier to be grounded (relative water depth $w \equiv D/H < 0.9$) and frozen to the bed. The numerical domain has a length of $L = 6 \cdot H \gg H$. The factor 6 was chosen as a compromise to reduce computational effort while ensuring that the upstream boundary does not effect stresses at the glacier terminus. $L$ could have been chosen to be truly "much larger" than $H$ but that would have required a lot of computation time without significantly benefiting the precision of the calculation. The flow-line assumption is justified, for example, in situations where the glacier is wide in comparison to its length and thickness. In these cases lateral stresses can often be neglected. The flow line assumption is a strong constraint which neglects, for example, any buttressing effects within the ice sheet. However, the considered geometry with the width of the glacier much larger than the horizontal extent in the flow-line direction $L = 6 \cdot H$ is

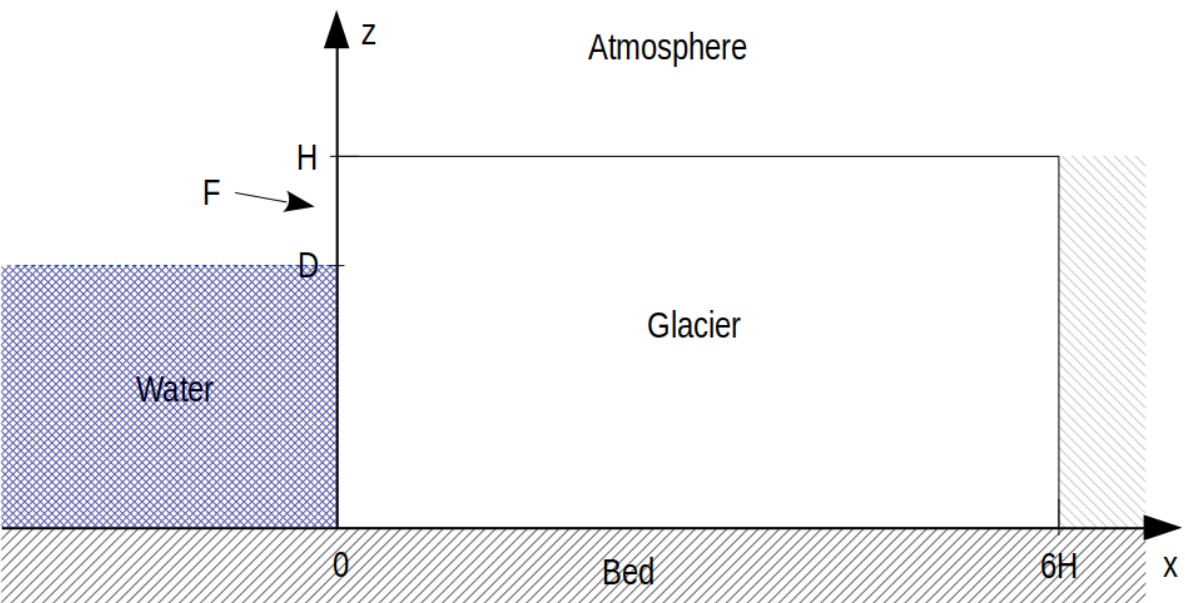

**Figure 1.** Geometrical set-up of the stress computation: two-dimensional plane flat glacier frozen to the bedrock with a calving front at its terminus. The glacier length $L$ is six times as large as the glacier height $H$ in order to ensure that the boundary condition on the right does not significantly influence the stress field at the terminus on the left. The ice thickness is denoted $H$, ice thickness below the water level is $D$ and the free-board is denoted $F$.

internally consistent and applicable to a number of situations observed both in Greenland and Antarctica. The assumption of a flat ice thickness is justifiable on a horizontal scale of several hundred meters to a few kilometers.

The ice flow and the stresses within the ice are governed by the Stokes equations,

$$\partial_x \sigma_{xx} + \partial_z \sigma_{xz} = 0, \tag{1}$$

5 $$\partial_x \sigma_{zx} + \partial_z \sigma_{zz} = f \tag{2}$$

and the continuity equation,

$$\nabla \cdot \boldsymbol{u} = \partial_x u_x + \partial_z u_z = 0 \tag{3}$$

with the Cauchy stress tensor $\sigma$ and the gravitational force $f$. The Cauchy stress tensor can be split into an isotropic pressure $P$ (also called cryostatic pressure) and the deviatoric stress tensor $S$, such that

10 $$\sigma_{ij} = -P \cdot \delta_{ij} + S_{ij} \tag{4}$$

where $\delta_{ij}$ is the Kronecker delta. Ice rheology is assumed to be given by Glen's flow law (van der Veen, 1999),

$$\dot{\epsilon}_{ij} = A S_e^{n-1} S_{ij}, \tag{5}$$

with the strain rate tensor $\dot{\epsilon}_{ij} = \frac{1}{2}\left(\partial_i u_j + \partial_j u_i\right)$ and the effective stress $S_e = \sqrt{\frac{1}{2}S_{xx}^2 + \frac{1}{2}S_{zz}^2 + S_{xz}^2}$.

The surface boundary is assumed to be traction-free. At the calving front boundary, we assume traction continuity to the water pressure and no traction above the water line. At the glacier bed, a no-slip boundary condition is assumed, which corresponds

5   to a glacier frozen to its bed. No inflow is assumed at the upstream boundary.

$$\text{ice top:} \qquad\qquad \sigma \cdot \boldsymbol{n} = \begin{pmatrix} \sigma_{xz} \\ \sigma_{zz} \end{pmatrix} = \mathbf{0}, \tag{6}$$

$$\text{ice base:} \qquad\qquad \boldsymbol{u} = \mathbf{0}, \tag{7}$$

$$\text{ice front:} \qquad\qquad \sigma \cdot \boldsymbol{n} = \begin{pmatrix} -\sigma_{xx} \\ -\sigma_{xz} \end{pmatrix} = \begin{cases} (-\rho_w g z, 0), & z < D \\ (0,0), & z > D \end{cases} \tag{8}$$

$$\text{upstream:} \qquad\qquad u_x = 0 \tag{9}$$

## 2.2   Numerical solution of the stress field

The boundary value problem was solved with the Finite Element package FEniCS (Alnæs et al., 2015) and stabilized with the Pressure Penalty method (Zhang et al., 2011). The numerical domain was divided into a regular triangular mesh with 100 vertical and 600 horizontal divisions.

Since the Stokes equation is linear in the stresses and the terminus boundary condition is linear in the ice thickness, the
equations can be solved on a dimensionless domain and the stresses scaled to arbitrary ice thickness. Velocities do not scale linearly but can be obtained from the scaled stresses through the ice rheology equation. The water depth at the calving front was incorporated via the relative (dimensionless) water depth $w = D/H$.

In order to determine a suitable stress-criterion for cliff calving we consider a number of commonly used stresses which
have a clear physical role (figure 2). Generally, stresses increase with ice thickness, while the presence of water at the glacier terminus decreases the stresses and stabilizes the calving front.

The deviatoric normal stress, $S_{xx}$, corresponds to an outwards force at the calving front which has two maxima, one at the waterline and one at the foot of the terminus. The deviatoric shear stress or Cauchy shear stress, ($S_{xz} = \sigma_{xz}$), translates to a bending moment which bends the top of the calving front forward and downward.

The different components of the deviatoric stress tensor are no invariants of the stress tensor, i.e. they depend on the coordinate system in which they are computed, and therefore they are not suitable as failure criteria. The largest principal stress,

$$\sigma_1 = \frac{\sigma_{xx} + \sigma_{zz}}{2} + \sqrt{\left(\frac{\sigma_{xx} - \sigma_{zz}}{2}\right)^2 + \sigma_{xz}^2}, \tag{10}$$

is calculated as the largest eigenvalue of the Cauchy stress tensor and corresponds to the largest normal stress in a given point. When $\sigma_1$ is positive, it is tensile and crevasses can form.

The maximum shear stress,

$$\tau_{max} = \sqrt{\left(\frac{\sigma_{xx} - \sigma_{zz}}{2}\right)^2 + \sigma_{xz}^2}, \tag{11}$$

acts on a plane at an angle $45°$ to the plane where the largest principal stress acts. It has its maximum at the foot of the calving front. The maximum shear stress can be related to brittle compressive failure (Schulson, 2001) and is therefore of particular interest for cliff failure.

The von Mises stress is the second invariant $J_2$ of the deviatoric stress tensor,

$$\sigma_{Mises} = \sqrt{\frac{3}{2}\left(S_{xx}^2 + S_{zz}^2 + 2S_{xz}^2\right)}, \tag{12}$$

and is used as a measure of deviatoric strain energy. It can also be related to material failure (Ford and Alexander, 1963) and has been used as a calving criterion by Morlighem et al. (2016). Since $S_{xx} = -S_{zz}$ due to the incompressibility of ice, the von Mises stress and the maximum shear stress differ only by a factor: $\sigma_{Mises} = \sqrt{3}\,\tau_{max}$.

## 3 Cliff failure criterion

In a first step we select a failure criterion which then yields a failure region based on the computed stress fields. In a second step we decide on a time scale for the failure in order to derive a simple calving law.

### 3.1 Partial thickness failure through crevasses

Crevasses are a natural candidate for ice front failure. In the case of glaciers that are frozen to the ground, crevasses, generally, do not form from the base upward (Ma et al., 2017). Instead, surface crevasses can form in the upper part of the glacier down to the depth where the principal stress becomes compressive, i.e. attains negative values (Nye, 1957). The presence of water at the calving front reduces the stresses in the ice and decreases the depth to which surface crevasses can penetrate. Surface crevasses, generally, do not penetrate through the whole glacier thickness and so crevasses cannot be the sole cause for calving. We thus do not follow this path to determine a failure region.

Surface meltwater filling surface crevasses can increase their depth (hydrofracturing) (Weertman, 1973; Das et al., 2008; Pollard et al., 2015), but this is also not considered here. The presence of crevasses weakens the ice and is expected to enable failure even when the critical shear stress is not yet exceeded but also this is not further considered here.

### 3.2 Full thickness shear failure

Instead, we assume shear faulting to be the dominant process in ice-cliff failure. We could use the von Mises stress as a failure criterion instead and reach qualitatively the same result, because they differ only by a factor of $\sqrt{3}$.

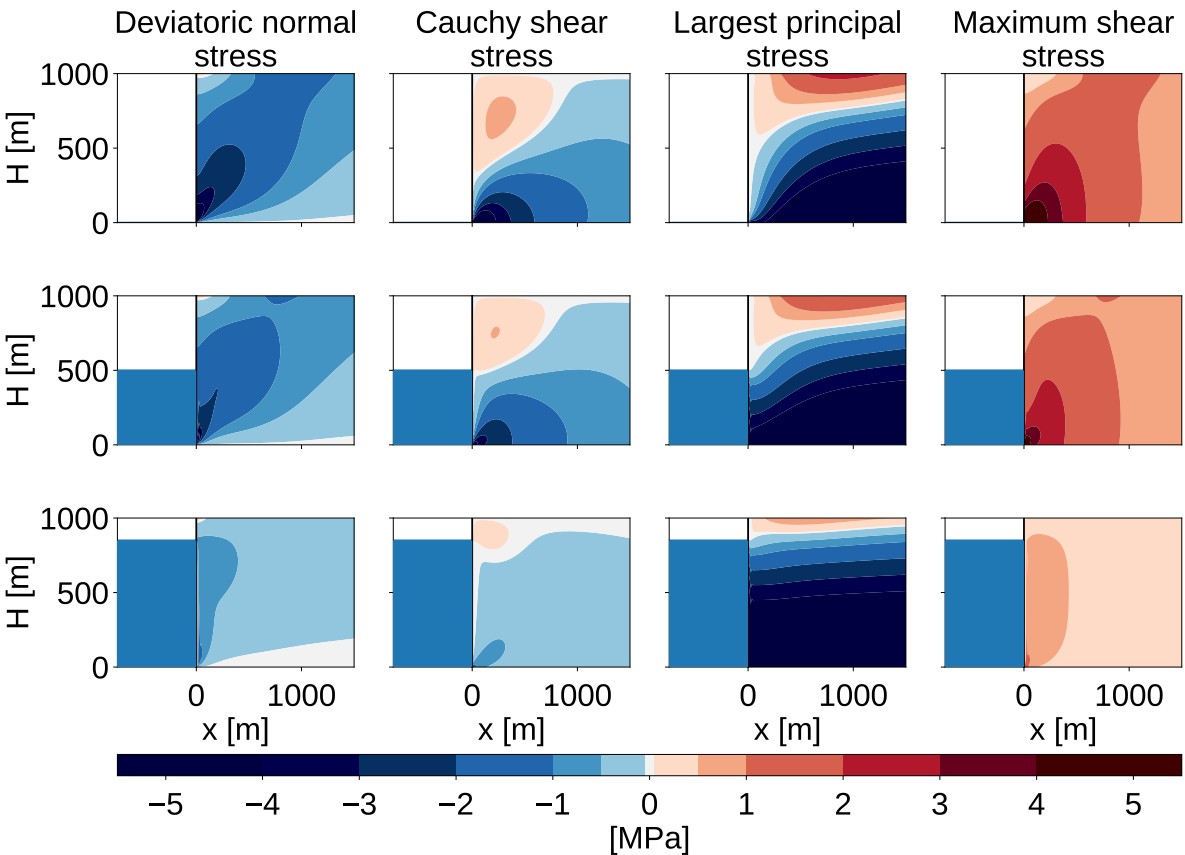

**Figure 2.** Stress configurations at the calving front for different relative water depths ($w = 0, 0.5, 0.85$) for a fixed ice thickness of 1000m. The first column shows the deviatoric normal stress in x-direction, $S_{xx}$, the second column shows the Cauchy shear stress, $\sigma_{xz} = S_{xz}$, the third column shows the largest principal stress, $\sigma_i$, and the last column shows the maximum shear stress, $\tau_{max}$.

The failure region is defined as the region close to the calving front where the maximum shear stress exceeds a critical shear stress of $\tau_c = 1\,\mathrm{MPa}$ (Schulson et al., 1999; Schulson, 2001). While the specific value of the critical shear stress may be subject to uncertainties (values might be between $0.5\,\mathrm{MPa}$ and $5\,\mathrm{MPa}$), it is mainly a constant that will not alter the calving rate dependence on the freeboard and the water depth. The specific choice of the value is motivated by laboratory experiments and can only provide an order of magnitude of the calving rate. However, the uncertainty resulting from this choice is smaller than the uncertainty arising from the estimate of the failure time (see below).

### 3.3 Comparison to Coulomb failure

In general, brittle compressive failure happens through shear faulting (Schulson et al. (1999)) and can be described with the Coulomb law (Weiss and Schulson (2009)): the shear stress $\tau$ acting on the future fault plane is resisted by material cohesion

$S_0$ and by friction $\mu\sigma$ with the friction coefficient $\mu$ and the normal stress across the failure plane $\sigma$. Failure happens, when:

$$\tau \geq S_0 + \mu\sigma \tag{13}$$

This expression depends on the direction of the fault plane. The failure condition can be expressed more generally in terms of the maximum shear stress $\tau_{max}$ and the isotropic pressure $P$ as

$$\sqrt{\mu^2 + 1}\,\tau_{max} = \tau_0 + \mu P \tag{14}$$

where $\tau_0$ is another measure of cohesive strength related to $S_0$ (Weiss and Schulson (2009)).

Weiss and Schulson (2009) provide values of $\mu = 0.3\ldots 0.8$ depending on the temperature of the ice. Since friction increases the strength of the ice, this could stabilize rather large ice cliffs. Bassis and Walker (2011) looked at upper bounds of glacier stability with a depth-averaged shear stress for different values of $\mu$ (0.65, 0.4, 0) and a cohesion of $\tau_0 = 1\,\mathrm{MPa}$. With a large friction coefficient, ice cliffs would be stable for freeboards of up to $600\,\mathrm{m}$ (see fig. 3) Since this is not observed in nature, they concluded that the best model is the one without friction which only allows freeboards of up to $200\,\mathrm{m}$. Thus with vanishing friction, the Coulomb failure criterion is equal to the maximum shear stress criterion used here.

## 4  Failure region

We define the failure region as the region close to the calving front where the maximum shear stress exceeds the critical shear stress $\tau_c$ anywhere in the ice column. The failure distance $L$ is the maximum distance of the failure region to the front and was determined for a range of ice thicknesses $H$ and relative water depths $w$ by solving the 2D Stokes equation numerically and tracing the contour line where the maximum shear stress $\tau_{max}$ equals the critical shear stress $\tau_c$ (see figure 4).

For a given water depth, the failure distance $L$ increases with the ice thickness $H$ or the glacier freeboard $F = H - D$ (figure 5). For glacier freeboards smaller than approximately $100\,\mathrm{m}$, the failure region vanishes: the critical shear stress is not exceeded anywhere in the ice and no shear failure takes place. This confirms results by Bassis and Walker (2011) which were derived analytically with some simplifications (see appendix A1 for more details). The relative water depths influences the slope of the freeboard - failure distance relation: for large relative water depths, the failure distance grows more quickly with increasing freeboard. This is because for a large relative water depth the overall ice thickness is much larger than for a similar freeboard with a smaller relative water depth and so the failure region is larger.

Above a critical freeboard of about $1000\,\mathrm{m}$ (see fig. 4 for $w = 0$ and $F = H$) the failure region encompasses the whole ice thickness. Below this critical value the failure region contains only the lower part of the ice thickness: but once the lower part of the ice column fails the upper part lacks support and fails as well. The freeboard - failure distance relation has a steeper slope for large freeboards when the whole ice thickness fails. This leads to a bend at the critical freeboard and hence the two parts require separate analytical fits. Here, we consider only values below the critical freeboard because that is the range of values most likely to occur in nature.

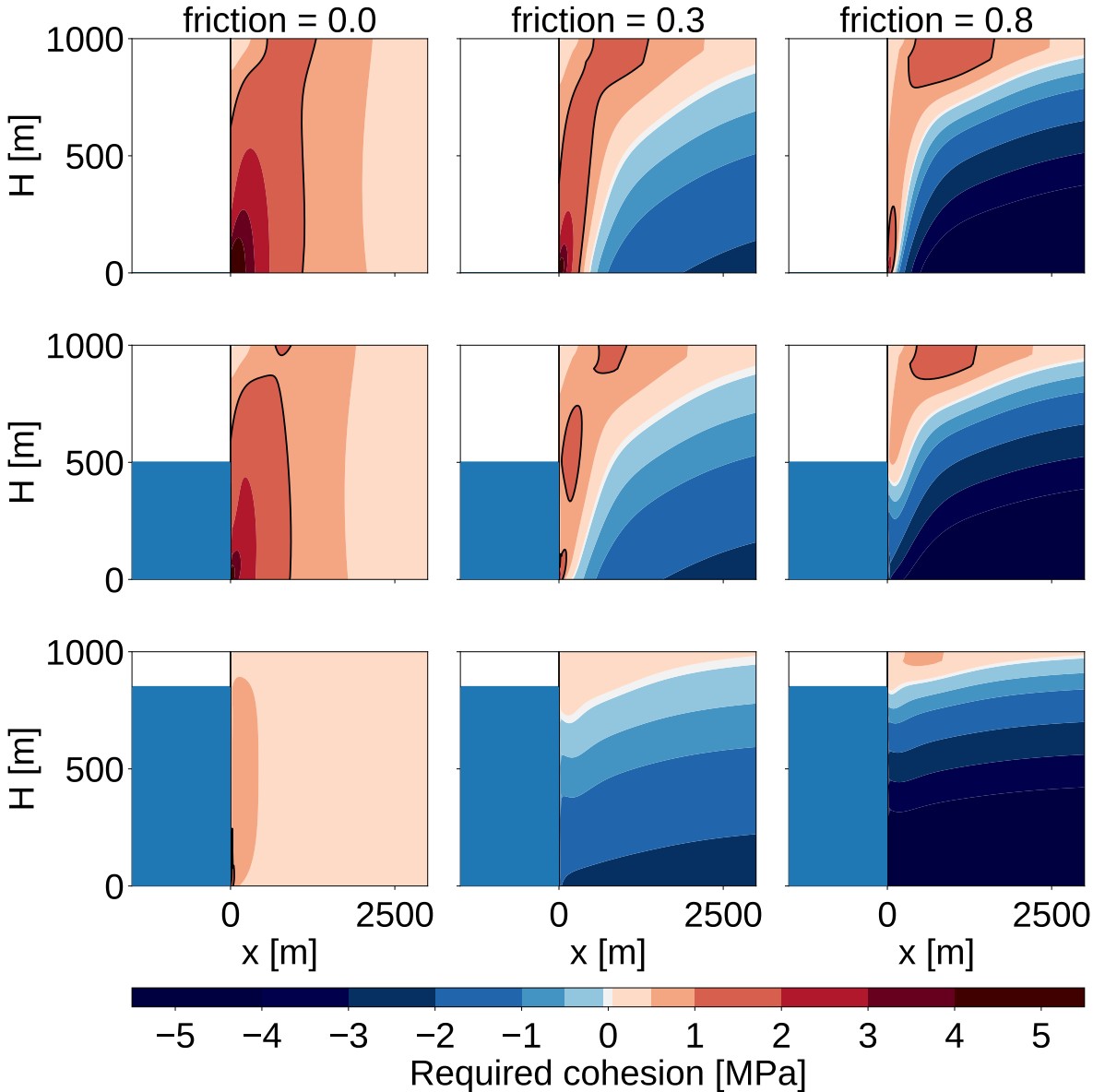

**Figure 3.** Assuming Coulomb failure, the required cohesion, $\tau_0 = \sqrt{\mu^2 + 1}\,\tau_{max} - \mu P$, is shown for different friction parameters ($\mu = 0, 0.3, 0.8$). The failure region for a maximum cohesion of $\tau_{max} = 1\,\mathrm{Mpa}$ is encased by the black line.

In figure 5 we provide an analytical fit with a power law function of the form

$$L = \left( \frac{F - F_c}{F_s} \right)^s \mathrm{m} \tag{15}$$

$$F_s = \left( 115 \cdot (w - 0.356)^4 + 21 \right) \mathrm{m} \tag{16}$$

$$F_c = (75 - w \cdot 49)\,\mathrm{m} \tag{17}$$

$$s = 0.17 \cdot 9.1^w + 1.76 \tag{18}$$

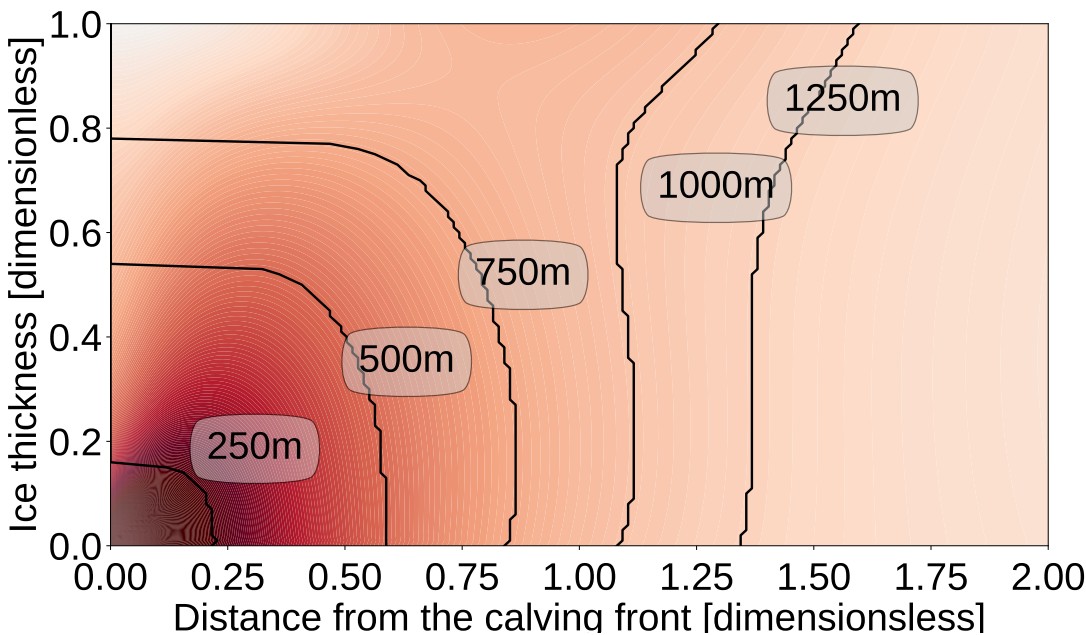

**Figure 4.** Outline of the failure region for different ice thicknesses on a dimensionless domain and without water stabilizing the front (ice thickness = glacier freeboard). The background color shows the maximum shear stress on a dimensionless scale with darker areas signifying larger stress. The failure region is defined as region close to the calving front where the maximum shear stress exceeds the critical shear stress $\tau_c$ anywhere in the ice column. The outline for $H = 1000\,\mathrm{m}$ is also shown in fig. 3 in the top-left panel.

with $w \equiv D/H < 0.9$ and $F \equiv H - D = H \cdot (1 - w)$. At first $L$ was fitted as a function of $F$ for each value of $w$. Then the parameter functions $F_s, F_c$ and $s$ were fitted as functions of $w$.

Fig. 5 shows the numerical results and the fit. Note that the fit has been optimized for relative error so for large freeboards the fit is a little off but it was considered more important to fit the onset of cliff calving correctly.

## 5 Failure time

There is a theory for damage evolution in ice for tensile damage (Pralong et al., 2003), from which the time to failure is derived as (Mercenier et al., 2018):

$$T_f = \frac{(1 - D_0)^{k+r+1} - (1 - D_c)^{k+r+1}}{(k + r + 1)B(\sigma_0 - \sigma_{th})^r} \tag{19}$$

with the rate factor for damage evolution $B$, material constants $r$ and $k$, initial damage $D_0$, critical damage $D_c$ and stress threshold for damage creation $\sigma_{th}$ and the working stress $\sigma_0$ which we assume to be the maximum shear stress $\tau_{max}$. With

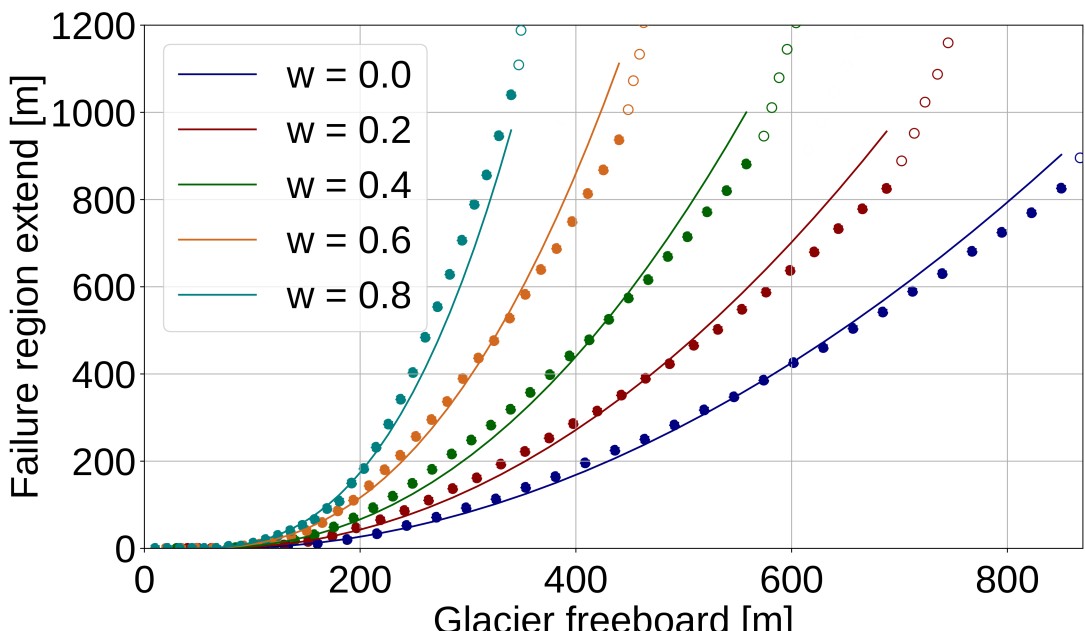

**Figure 5.** Size of shear failure region $L$ as a function of glacier freeboard $F = H - D$ and relative water depth $w = D/H$. Numerical results are shown for smaller freeboards where the failure region does not encompass the whole ice thickness (filled dots) and for large freeboards, where the failure region contains the whole ice thickness (empty circles). A power law has been fitted to the numerical results for small freeboards (continuous line), which is given by eq. 15. The fit has been optimized for relative error in order to get the onset of cliff calving right.

these assumptions eq. 19 can be written as

$$T_f = \cdot(\sigma_0 - \sigma_{th})^{-r}/B^* \tag{20}$$

with $\sigma_{thr} = 0.17\,\mathrm{MPa}$, $r = 0.43$ and $B^* = 65\mathrm{Mpa}^{-r}\mathrm{a}^{-1}$, as given in Mercenier et al. (2018). These parameters have been determined by calibrating a tensile failure calving model with data on calving rate, water depth and ice thickness for a variety

5   of tidewater glaciers in the Arctic.

However, eq. 20 is valid only for damages created through tensile creep. The difference between tensile and compressive damage is that under tension a single crack grows in an unstable fashion to cause failure while in compression a large number of small cracks grows in a stable fashion until their interaction causes failure (Ashby and Sammis, 1990).

There is plenty of literature about compressive creep and failure in rocks (Brantut et al., 2013). Fatigue failure happens when

10   a material is loaded with stresses below the failure stress and fails with a time delay due to the development of micro cracks.

There is an exponential law as well as a power law for the time to failure:

$$t_f = t_0 \exp\left(-b\frac{\sigma}{\sigma_0}\right) \tag{21}$$

$$t_f = t'_0 \left(\frac{\sigma}{\sigma_0}\right)^{-b'} \tag{22}$$

The power law exponent is usually large, $b' \approx 20$, so the power law is very similar to the exponential law. Once the major stress $\sigma$ exceeds the instantaneous strength $\sigma_0$, immediate failure is assumed ($t_f = 0$). Both time to failure relations fit the experimental data for rock well (Amitrano and Helmstetter, 2006). However, the constants depend on material properties and there are to our knowledge no studies for time dependence of compressive creep failure in ice.

This leaves us with a dilemma: There have been no studies that determined the material properties of ice under time-dependent brittle compressive failure. Also, we cannot determine those material properties ourselves by fitting the resulting calving law to observations, because so far cliff calving has not been observed as the major calving process in any glacier. That makes it impossible to estimate the time to failure using eq. 21 or 22. Eq. 20 and the value of its constants have been determined for tensile failure, which is microscopically very different from brittle compressive failure. So there is little reason to expect it to describe the timescale of shear failure well.

Nevertheless, we will use it as a starting point for our further analysis: For the stresses above the shear failure threshold, $\sigma_0 > 1\,\text{MPa}$, the time to failure for tensile failure (given by eq. 20) changes by only a factor of 2 (see fig. 6). Hence, the calving relation can be further simplified by assuming that there is a characteristic time to failure, $T_c$, that is the same for all stresses and sizes of failure regions, $T_c \approx 4\,\text{days}$. This characteristic time has been derived from parameters determined for tensile failure, so its application to shear failure comes with an uncertainty that is is difficult to quantify.

## 6   Calving law

With a constant failure time, the calving rate is proportional to the size of the failure region

$$C = C_0 \cdot \left(\frac{F - F_c}{F_s}\right)^s \tag{23}$$

$$F_s = \left(115 \cdot (w - 0.356)^4 + 21\right)\,\text{m} \tag{24}$$

$$F_c = (75 - w \cdot 49)\,\text{m} \tag{25}$$

$$s = 0.17 \cdot 9.1^w + 1.76 \tag{26}$$

$$C_0 = \frac{1\,\text{m}}{4\,\text{days}} = 91.25\,\text{m/a} \tag{27}$$

with $w \equiv D/H < 0.9$ and $F \equiv H - D = H \cdot (1 - w)$.

A dry cliff ($w = 0$) reaches calving rates of $C = 50\,\text{km/a}$ at an ice thickness of $F = H \approx 800\,\text{m}$, while an ice cliff that is close to floatation ($w = 0.8$) reaches the same calving rate at a freeboard of $F \approx 300\,\text{m}$, which corresponds to an ice thickness of $H \approx 1500\,\text{m}$ (see fig. 7).

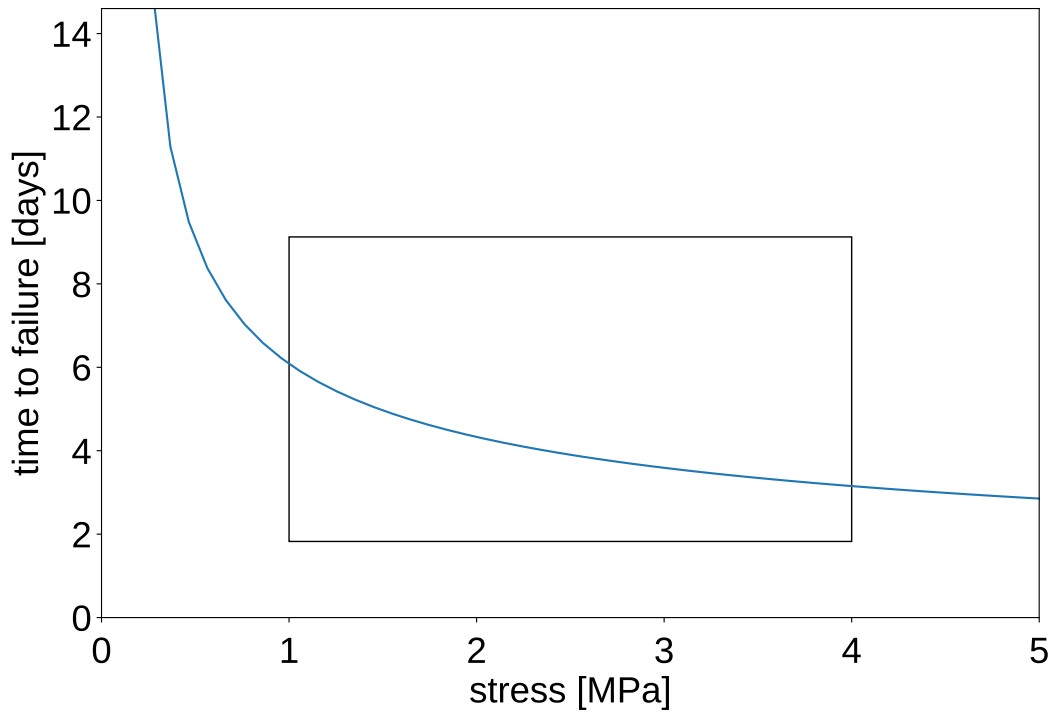

**Figure 6.** Time to failure given by eq. 20. For stresses above the shear failure threshold, $\sigma_0 > 1\,\mathrm{MPa}$, the time to failure changes only little (box).

How do cliff calving rates given by eq. 23 compare to currently observed calving rates? A glacier enters the cliff calving regime, when its freeboard is larger than the critical freeboard $F_c$ and the cliff calving rate given by eq. 23 becomes nonzero. Obviously, glaciers calve through tensile failure before and after they reach the cliff calving regime, so we expect the overall calving rate to be larger than the cliff calving rate, especially for glaciers that just entered the cliff calving regime and are
5  heavily crevassed.

Jakobshavn glacier in Greenland is one of the few glacier that are currently in a cliff calving mode. Jakobshavn glacier terminates in water with a depth of $800\,\mathrm{m}$ (Morlighem et al., 2014) and has a glacier freeboard of $100\,\mathrm{m}$ (Xie et al., 2018). Therefore it can be considered to be at the beginning of the cliff calving regime. Since the terminus is also heavily crevassed, we expect tensile calving to be the main contribution to the overall calving rate. Hence, this example can only give an upper
10  bound on the possible cliff calving rate.

It is difficult to determine calving rates directly. The ice flow velocity to the front of Jakobshavn is up to $12\,\mathrm{km/a}$ (Morlighem et al., 2014). The grounding line of Jakobshaven glacier retreats and advances seasonally about $6\,\mathrm{km}$ per year, but the maximum

| $w$ | $s$ | $F_c$ | $F_s$ |
|-----|------|------|-------|
| 0 | 1.93 | 75 | 22.85 |
| 0.1 | 1.97 | 70.1 | 21.49 |
| 0.2 | 2.02 | 65.2 | 21.07 |
| 0.3 | 2.09 | 60.3 | 21.00 |
| 0.4 | 2.17 | 55.4 | 21.00 |
| 0.5 | 2.27 | 50.5 | 21.05 |
| 0.6 | 2.40 | 45.6 | 21.41 |
| 0.7 | 2.56 | 40.7 | 22.61 |
| 0.8 | 2.75 | 35.8 | 25.47 |
| 0.9 | 3.00 | 30.9 | 31.07 |

**Table 1.** Table of parameters in the cliff calving relation eq. 23, giving the exponent $s$, critical freeboard $F_c$ and scaling factor $F_s$ for a range of relative water depth values $w$.

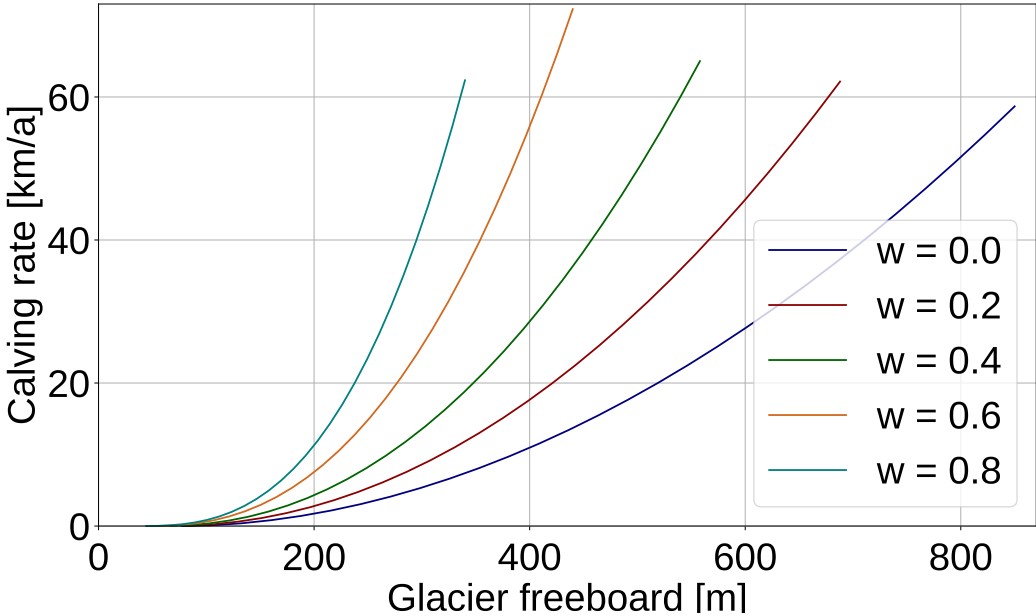

**Figure 7.** Cliff calving rates $C$ as a function of glacier freeboard $F = H - D$ and relative water depth $w = D/H$ as given by eq. 23.

grounding line position has not changed much between 2012 and 2015 (Xie et al., 2018). Assuming a fixed grounding line, the calving rate would equal the flow velocity. Hence the averaged yearly calving rate is approximately $12\,\mathrm{km/a}$.

Inserting values of glacier freeboard and water depth given above into eq. 23 gives a cliff calving rate of $C = 750\,\mathrm{m/a}$, which is well below the overall calving rate.

## 7 Discussion and Conclusion

We solved the 2D Stokes equation numerically for a flat glacier frozen to its bed in a flow-line model and investigated the stresses at the calving front.

Four simplifications were made:

1. The model was solved in one horizontal direction, neglecting lateral shear effects. Without lateral shear effects, the result is independent of the topography of individual glaciers.

2. We assumed a basal boundary condition corresponding to a glacier frozen to its bed. Sliding was not considered.

3. The main failure mechanism was assumed to be shear faulting. We assumed brittle compressive failure according to the Coulomb law without friction stabilizing the ice cliff. Friction would allow glaciers with larger freeboards than observed to be stable.

4. A constant time to failure has been assumed.

Under these assumptions, crevasses cannot penetrate the whole glacier depth and shear failure was chosen as the main failure mechanism. The region where shear stresses exceed a critical shear stress of $1\,\mathrm{MPa}$ is called the failure region. The extend of this failure region, the failure distance, was determined for a range of glacier freeboards and relative water depths. For freeboards small enough for the failure region not to encompass the whole ice thickness, an analytical fit was made. Assuming a constant time to failure, a cliff calving rate was derived. Resulting cliff calving rates seem large compared to currently observed calving rates. Comparison with Jakobshavn glacier in Greenland shows that the cliff calving rate is smaller than the overall calving rate, hence we conclude that eq. 23 probably does not overestimate cliff calving rates.

### 7.1 Idealized setup vs. realistic conditions

The cliff calving rate was derived using an idealized setup, given by the first two of the four assumptions described above. Realistic glaciers that might experience cliff calving sit in valleys where they experience lateral drag and may be sliding. The calving front may have a slope rather than a vertical cliff and there might be an undercut caused by frontal melt.

### 7.1.1 Sliding glaciers

First consider sliding with a constant velocity $v$ (i.e. vanishing strain rate) for which the upstream boundary condition is an influx with velocity $v$, so $u = v$. The basal boundary conditions become $u = v, w = 0$. Solving the Stokes' equations with these boundary conditions numerically with FeniCS gives the exact same stress fields as in the frozen case and the velocity field is

simply shifted by the sliding velocity $v$. This is not surprising: A simple Galilean transformation takes this sliding glacier back to the frozen glacier previously considered without changing any of the physics.

In general, sliding velocities increase towards the glacier terminus. The steepest possible velocity gradient can be obtained with a free-slip basal boundary condition: we assume no influx at the upstream boundary, $u = 0$, and at the bed we assume free

slip in the horizontal direction, which only leaves a boundary condition for the vertical velocity, $w = 0$. The basal velocity is zero at the upstream boundary and takes its maximum at the calving front. Due to this velocity gradient, the maximum shear stress is large throughout the whole numerical domain (see fig. 8). For increasing ice thickness it becomes difficult to define a meaningful failure region, because the critical shear stress is exceeded in the whole numerical domain - one must assume that the whole numerical domain will fail. Thus, in the case of a sliding glacier, the failure region is larger than in the case of a

glacier frozen to its bed. Hence, the derived cliff calving rate can serve as a lower bound for this kind of calving fronts.

To summarize: The derived cliff calving law is valid for glaciers that are frozen to the bed or sliding with a constant velocity and vanishing strain rate. It serves as a lower bound on the calving rate for glaciers in which velocities increase towards the calving front.

### 7.1.2    Lateral drag

In order to investigate how lateral drag influences cliff calving, we will assume ice flow in a channel with a flow-line in the x-direction. Ice is assumed to flow only in the x-direction with a flow maximum in the middle of the channel. Since deviatoric stresses are connected to the strain rate, $\tau_{ij} = B\dot{\epsilon}_e\dot{\epsilon}_{ij}$, and the strain rate is given by the velocity gradients, $\dot{\epsilon}_{ij} = \frac{1}{2}(\partial_i u_j + \partial_i u_j)$, we get an additional deviatoric shear stress in the x-y-plane, $\tau_{xy}$. The other stress components in y vanish, $\tau_{yz} = \tau_{yy} = 0$, because the respective velocity gradients vanish. The Cauchy stress tensor becomes

$$\sigma = \begin{pmatrix} P + \tau_{xx} & \tau_{xy} & \tau_{xz} \\ \tau_{xy} & P & 0 \\ \tau_{xz} & 0 & P - \tau_{xx} \end{pmatrix} \tag{28}$$

The principal stresses $\sigma_i$ are defined as eigenvalues of $\sigma$, and the maximum shear stress $\tau_{max}$ as the the difference between the maximum and minimum principal stress. In 3D, there is no simple analytical formula for the eigenvalues of a matrix and therefore it is not feasible to get an analytical estimate on whether the introduction of non-zero $\tau_{xy}$ makes $\tau_{max}$ smaller or larger.

Assuming $P(x,z)$, $\tau_{xx}(x,z)$ and $\tau_{xz}(x,z)$ as given by the FeniCS simulation with a constant $\tau_{xy} = 1\,\mathrm{MPa}$, we calculate the principal stresses and the maximum shear stress numerically. This shows that $\tau_{max}$ increases with increasing absolute value of $\tau_{xy}$ (see fig. 9).

Hence, lateral shear increases the maximum shear, therefore increasing the size of the failure region and the cliff calving rate. The derived cliff calving rate can serve as a lower bound if lateral drag is present.

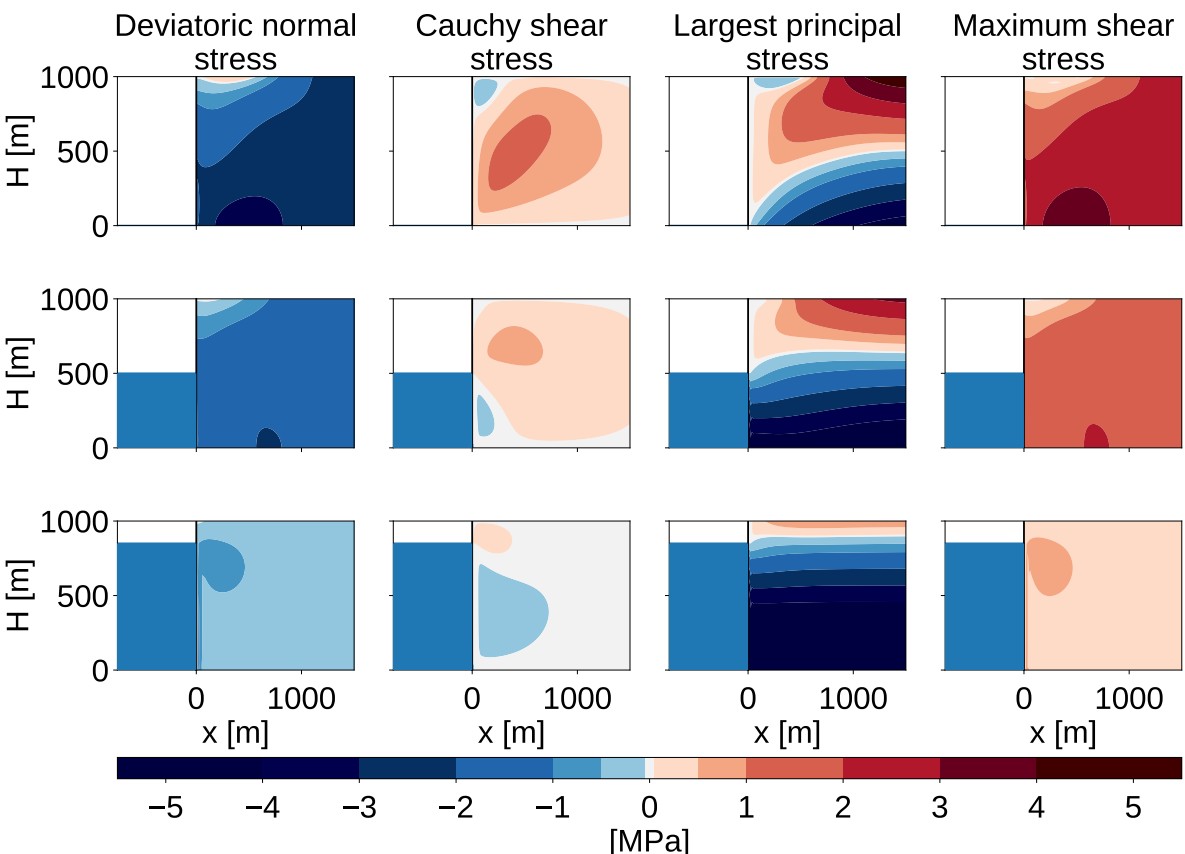

**Figure 8.** Stress configurations at the calving front for different relative water depths ($w = 0, 0.5, 0.85$) for a fixed ice thickness of 1000m with a free-slip basal boundary condition instead of the no-slip boundary condition used in the previous analysis (compare fig. 2). The first column shows the deviatoric normal stress in x-direction, $S_{xx}$, the second column shows the Cauchy shear stress, $\sigma_{xz} = S_{xz}$, the third column shows the largest principal stress, $\sigma_i$, and the last column shows the maximum shear stress, $\tau_{max}$. In contrast with the no-slip case, there is no definite failure region as the maximum shear stress is large throughout the whole numerical domain.

### 7.1.3 Calving front slope

Other studies have shown that a calving front with a slope has significantly reduced stresses compared to a calving front with a vertical cliff (Benn et al., 2017; Mercenier et al., 2018). It is clear that a calving front slope would also reduce the cliff calving rate.

5    We have not analyzed this effect here, because once cliff calving has been initiated, the full thickness calving probably prevents calving front slopes from forming. We aim to find a parametrization that can be implemented in ice sheet models capable of simulating the Antarctic ice sheet. These simulations are done on resolutions of several kilometers and cannot resolve calving front slopes on length scales of several tens or hundreds of meters.

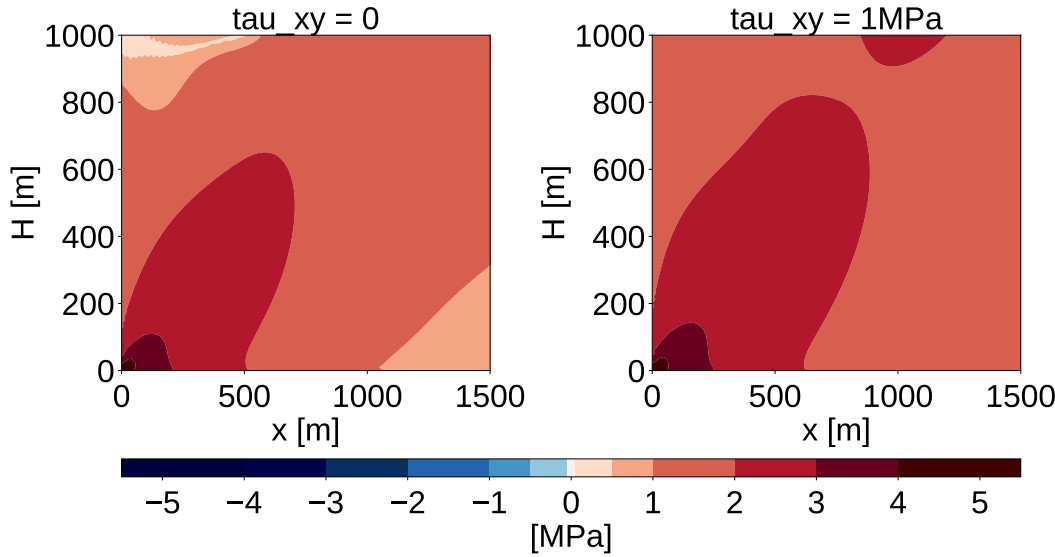

**Figure 9.** Maximum shear stress $\tau_{max}$ in the vicinity of the calving front in the case without lateral drag (left) and with a constant lateral drag of $\tau_{xy} = 1\,\mathrm{MPa}$ (right).

### 7.1.4 Melt-undercut

Undercut from melt would increase the stresses near the calving front (Benn et al., 2017) and hence increase the calving rate.

### 7.2 Uncertainties

Cliff calving is still a rather hypothetical process with a very limited scope of observations. Since there are currently no glaciers
that are clearly in a cliff calving regime, the calving rate cannot be fitted to observed calving rates. There is uncertainty in the
maximum shear stress used to determine the failure distance as well as the time to failure.

Laboratory studies give a range of values between $0.5\,\mathrm{MPa}$ and $5\,\mathrm{MPa}$ for the critical shear stress (Schulson et al., 1999;
Schulson, 2001) . A much larger uncertainty arises from the time to failure. There are studies that give time to failure relations
and parameters for brittle compressive of rocks, but none for ice. Time to failure of ice has only been studied for tensile failure.
We use the time to failure relation used by Mercenier et al. (2018) as a first guess. Applying this time to failure for tensile
failure to a process of shear failure is very uncertain. We guess that the time to failure could be up to an order of magnitude
smaller or larger.

The scaling parameter $C_0$ in eq. 23 should therefore be considered a free parameter. In any implementation of this cliff
calving relation, a range of values for $C_0$ should be tested for plausibility.

### 7.3 Comparison with other calving parametrisations

#### 7.3.1 Other cliff calving approaches

Bassis and Walker (2011) derived a stability limit for ice cliffs considering shear and tensile failure (their assumptions are analyzed further in the appendix). According to eq. 23, cliff calving starts when the freeboard exceeds $F \approx 75\,\mathrm{m}$, this is close to the stability limit of $F \approx 100\,\mathrm{m}$ given by Bassis and Walker (2011).

Pollard et al. (2015) and DeConto and Pollard (2016) implemented cliff calving in their ice sheet model by assuming a cliff calving rate that is zero until the freeboard has reached $\approx 100\,\mathrm{m}$, increases linearly up to $3\,\mathrm{km/a}$ for a freeboard of about $150\,\mathrm{m}$ and stays constant after that. The calving relation is modified by factors representing back stress and additional wet-crevasse deepening. Edwards et al. (2019) did an ensemble study with a range of values for the maximum cliff calving rate from $0\,\mathrm{km/a}$ (no cliff calving) up to $5\,\mathrm{km/a}$. Depending on the scaling constant $C_0$, cliff calving rates given by eq. 23 have an equal range of magnitude, but increase with a power-law dependence and have no upper bound.

Bassis et al. (2017) implemented cliff calving by requiring that ice cliffs cannot exceed the stability limit. This becomes a condition for the speed of grounding line retreat and advance. Eq. 23 is easier to implement in ice sheet models, because it can be implemented just like other calving parametrizations and does not need to be rewritten as a condition for the grounding line.

#### 7.3.2 Other stress-based calving laws

Mercenier et al. (2018) derived a cliff calving law for tidewater glaciers below the stability limit by solving the stresses in the vicinity of the front and assuming tensile failure through the formation of a large crevasse. In contrast, we assume shear failure (also called brittle compressive failure). The calving rate given by Mercenier et al. (2018) increases approximately linearly with the freeboard and has no lower bound, while the calving rate given by eq. 23 grows with a power $s(w) > 1$ for freeboards larger than the critical freeboard $F_c(w)$ (see fig. 10). Hence, we expect tensile failure to dominate for small freeboards and shear failure to dominate for large freeboards.

It is difficult to say at which glacier freeboard the tensile failure regime ends and the shear failure regime begins, not only due to uncertainty in the scaling parameter $C_0$. In practice, both failure modes will interact, with tensile stress damaging the ice through few large crevasses originating from the surface of the ice and shear stress damaging the ice through a large number of small crevasses in the lower part of the cliff. This likely interaction of failure modes cannot be analyzed by assuming ice to be a continuous medium (like the approach used here and by Mercenier et al. (2018)), but should be done with damage theory or a discrete element approach.

### 7.4 Conclusion

The calving law proposed here was derived under a number of constraining assumptions. First it was assumed that friction plays no role in shear failure. Secondly it was assumed that once the critical shear stress is exceeded, ice fails after a constant time to failure. An improved cliff calving model might include friction and allow a stress-dependent time to failure.

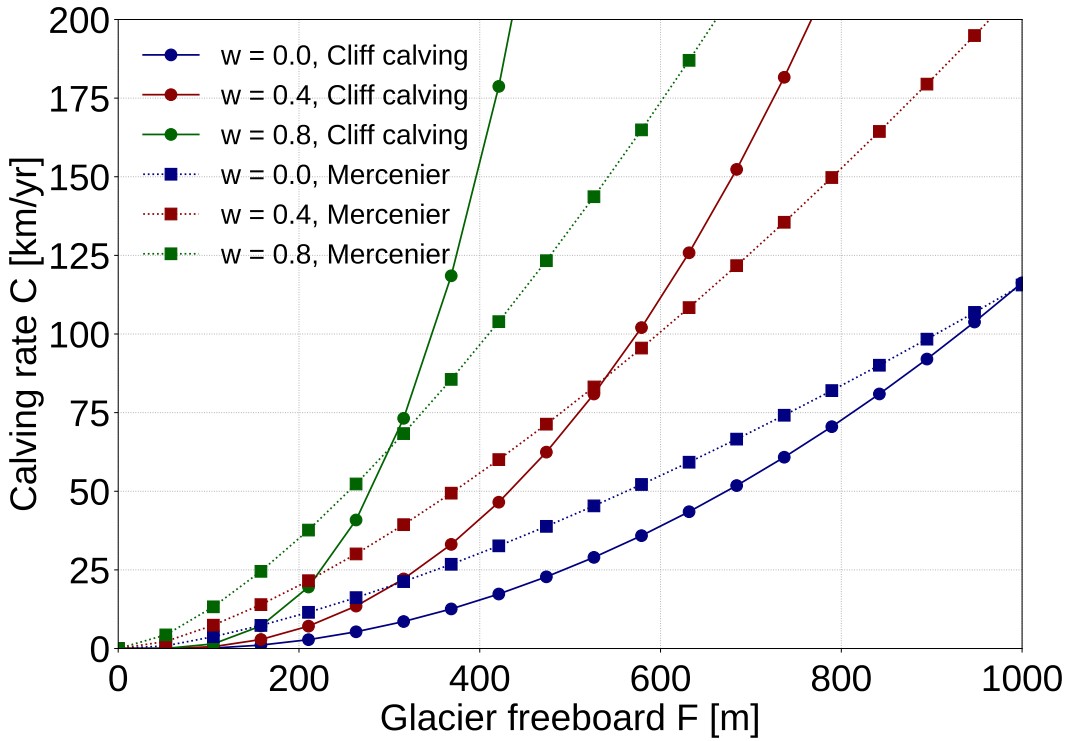

**Figure 10.** Comparison of the cliff calving law given by eq. 23 (continuous line) with the calving law for tidewater glaciers given by Mercenier et al. (2018), eq. 22 (dotted line). Note that the cliff calving rate could be scaled differently due to the uncertainty in $C_0$

.

If the Coulomb law with a friction component is used, the immediate failure region is smaller than in the no-friction case. Time to failure relations for compressive failure as given by 21 and 22 are valid for stresses below the critical shear stress. Failure is assumed to be instantaneous as soon as the critical shear stress is reached. Regions where the stress is below the failure stress would be assigned a stress-dependent failure time leading to a spatially distributed time to failure. Since friction is smaller at the top of the ice cliff, the top would fail earlier than the base, leaving a foot that would subsequently fail due to buoyant forces. There is no simple way to find a parametrization of the cliff calving rate for these processes.

Another problem is that there are no laboratory studies on the parameters in the time to failure relations for ice. It is also not possible to calibrate the calving relation using observed calving rates, because there are no glaciers currently available where cliff calving is the primary failure mechanism. Paleorecords might provide some means to calibrate cliff-calving rates as attempted in Pollard et al. (2015) and DeConto and Pollard (2016).

Paleorecords might not be constraining enough to provide a useful limit for the Antarctic sea level contribution of the next 85 years. But even if it is difficult to constrain the rate of cliff-calving there are important qualitative consequences of a monotonously increasing cliff-calving dependence on ice thickness. The most important is the potential of a self-amplifying

ice loss mechanism which is not constraint by the reduction in calving but must be constraint by other processes. Without some kind of cliff-calving mechanism it is likely that ice sheet models are lacking an important ice loss mechanism.

*Code availability.* FeniCS can be downloaded from the project website https://fenicsproject.org/download/. The script used for the FeniCS simulation in this paper is available on request from the authors.

## 5 Appendix A: Simplified stress balance

It is possible to solve the stress balance at the calving front analytically in a depth-averaged model with a simplifying assumption for the isotropic pressure. This has been used by (Bassis and Walker, 2011) and (Pollard et al., 2015). It is interesting to compare this with the numerical stress field solution obtained above.

(Bassis and Walker, 2011) and (Pollard et al., 2015) assumed the isotropic pressure is given by the gravitational pressure

$$P(x,z) = \rho_i g (H - z) \tag{A1}$$

where $\rho_i$ is the density of ice. This assumption is actually only true over length scales that are large compared with the ice thickness and far from the ice margins (MacAyeal, 1989), which is not the case when stresses close to the calving front are analyzed. But making this assumption allows for an analytical solution of the depth-averaged stresses and does not require any ice rheology.

Together with incompressibility, which means that the trace of the strain rate disappears ($\dot{\epsilon}_{kk} = 0$) and implies $S_{xx} + S_{zz} = 0$, the 2D Stokes equations become:

$$0 = \frac{\partial S_{xx}}{\partial x} + \frac{\partial S_{xz}}{\partial z}, \tag{A2}$$

$$0 = \frac{\partial S_{xz}}{\partial x} - \frac{\partial S_{xx}}{\partial z}. \tag{A3}$$

Assuming a traction-free surface boundary, traction-continuity at the terminus boundary and vanishing deviatoric stresses at the upstream boundary as well as the bed boundary, a boundary value problem arises that can be solved numerically.

The resulting stresses are smaller than the stresses obtained in section 2 for the 2D Stokes equation with nonlinear ice rheology (figure A1). A failure region can be defined as in section 3 and its size shows a very similar dependence on glacier freeboard and water depth, though it is smaller by about a factor of three.

The biggest difference between the two approaches lies in the largest principal stress: In this simplified problem, the largest principal stress is negative in the whole ice volume; there is no region of tensile stresses, so no crevasses form. This is due to the assumption that the isotropic pressure is equal to the gravitational pressure, which is not actually the case in the vicinity of the glacier terminus.

*Competing interests.* The authors declare no competing interests.

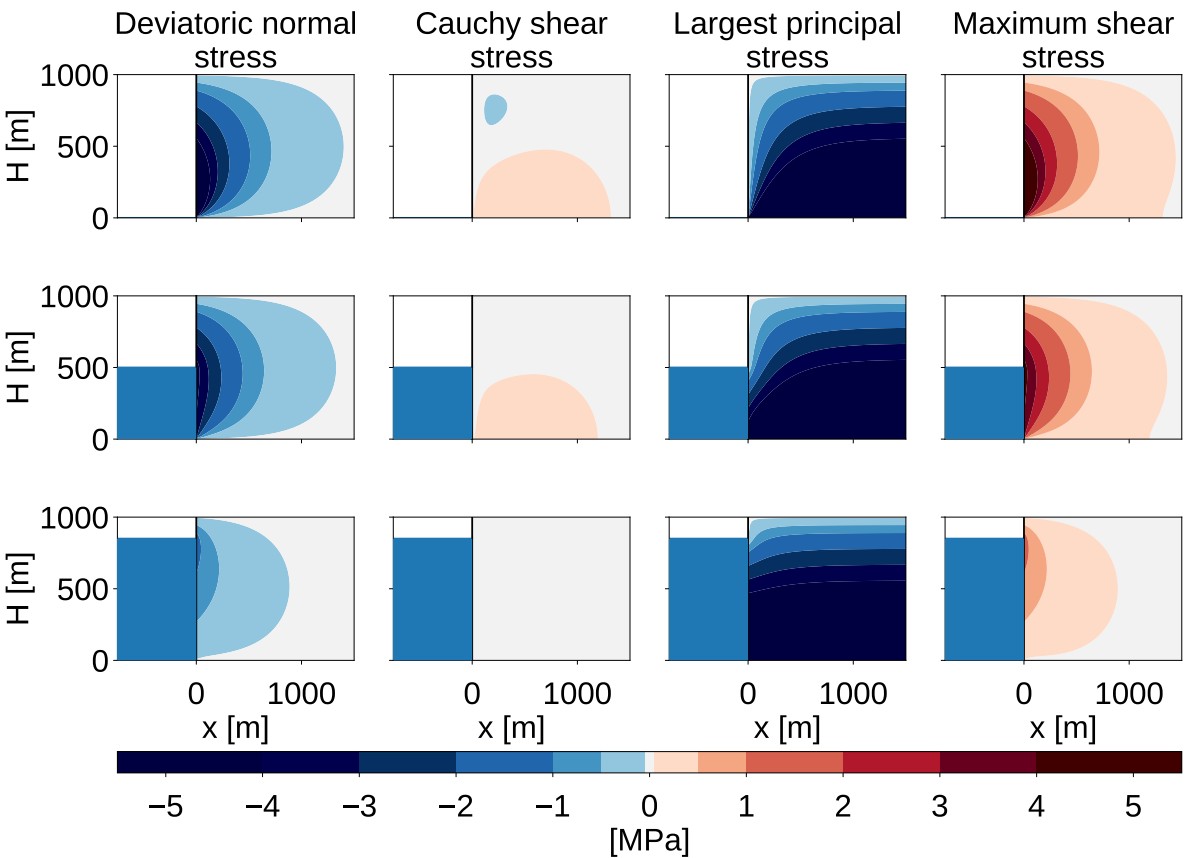

**Figure A1.** Stress configurations at the calving front for different relative water depths ($w = 0, 0.5, 0.85$) for a fixed ice thickness of 1000m. The first row shows the deviatoric normal stress in x-direction, $S_{xx}$, the second row shows the Cauchy shear stress, $\sigma_{xz} = S_{xz}$, the third row shows the largest principal stress, $\sigma_i$, and the last row shows the maximum shear stress, $\tau_{max}$.

*Acknowledgements.* TS was funded by the Heinrich Böll Foundation. TS would like to thank Yue Ma and Christian Helanow for their valuable help with FeniCS.

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
