# Peer review of "A simple stress-based cliff-calving law"

_The Cryosphere, 2018_

## Referee Comment (RC1) · Anonymous Referee #1 · 9 Nov 2018

This paper presents a new calving law designed to calculate calving rates from unstable cliff-like caving fronts. The authors use a two-dimensional flowline geometry to calculate maximum shear stresses in the ice. A critical shear stress is then used to define a region of failure - the ice which will be lost in a calving event. An analytical fit to the results is used to develop a generalised formula relating calving length to freeboard and relative water depth, and the authors then use a constant failure time of the ice to convert this to a calving rate.

This paper will be of interest to the community, as it provides a new method of calculating calving rates for unstable ice cliffs. This is a topic of considerable interest, as calving models which include cliff failure produce substantially different sea level rise predictions than other ice sheet models (e.g. DeConto and Pollard, 2016). A calving

law of this nature will always require simplifications, and I appreciate the authors have taken time to be clear about the assumptions made. However, these assumptions are significant and are likely to be highly limiting in when the calving law could or should be applied. I would like to see the authors comment further on the conditions under which this calving law would be valid and how widely they occur. I strongly recommend expanding the discussion of the assumptions made, so readers fully appreciate when and where the model can reasonably be used. There are also a number of errors and omissions in the paper which should be corrected. A list of specific recommendations is provided below.

**Major comments**

The model used to derive the proposed analytical calving law has a highly simplified geometry – with zero surface or bed slope, no lateral drag, and no sliding at the bed. Previous studies have shown that the extent of the "failure region" discussed in this paper is strongly affected by basal sliding rates (Ma et al., 2017). Likewise, other studies have shown that the stress regime around the calving front is strongly affected by surface slope (Mercenier et al, 2018).

As the authors point out, "there are no glaciers currently available where cliff calving is the primary failure mechanism", but modelling studies such as DeConto and Pollard (2016) suggest cliff failure could occur in future in deep Antarctic basins, after rapid retreat of their buttressing ice shelves. These environments are highly likely to experience basal sliding, as well as lateral drag. It is hard to say what proportion of ice cliffs might meet the authors' conditions, but the proposed model for predicting calving rates seems a lot less generally applicable than simply using the maximum shear stress to define a new calving front location. At the very least the paper should include more discussion of precisely what circumstances the model is valid for, and under what conditions (e.g. basal sliding) it is likely to fail.

**Minor comments**

Page 2 line 6: Columbia glacier is in Alaska, not Canada

Page 2 lines 28-29: The description of Mercenier et al. (2018) is extremely brief and doesn't contrast the model with other studies, which would be much more informative. This also seems a suitable point to reference Morlighem et al. (2016).

Introduction: The introduction misses damage mechanics methods which have been used to implement calving in a tidewater glacier (Krug et al., 2014).

Page 3 line 5: "it is not clear what a cliff calving law would look like". Are the authors aware of Bassis et al. (2017) which already implemented a calving law based on cliff instability?

Page 4 lines 5-10: No boundary condition is provided for the upstream boundary of the model (r.h.s. in figure 1)

Page 5, eq. 11: Should y in this equation be z?

Page 7 lines 2-3: "However, it does not take into account whether deviatoric stresses are tensile or compressive or shear stresses and this is likely to be important for ice failure." Surely this is the advantage of using the von Mises stress as a criterion – it is able to allow for failure under both tension and shear, and is therefore more widely applicable than a criterion that considers only one mechanism of failure?

Page 7 lines 5-10: these uncertainties should be explored further in the discussion, which doesn't currently make their magnitude clear.

Page 7 eq. 13: I'm not sure where the term  $sqrt(\mu^2+1)$  on the l.h.s. comes from here.

Page 9 line 1: "Above a critical freeboard of about 1000m the failure region encompasses the whole ice thickness." Is this based on results from figure 4?

Page 9 lines 3-4: "The freeobard [sic]- failure region relation has a bend at the critical freeboard and hence the two parts require separate analytical fits" Figure 5 shows no freeboards above 800 m, so readers cannot see how this conclusion was reached.

СЗ

Page 10 eq. 18: What are k, r and B?

Page 11 eq. 19 & 20: I think  $\sigma$  and  $\sigma$ 0 here are not the same as in previous equations?

Page 11 eq. 21: Is part of this equation missing? What values have you used for k, D0 and Dc?

Page 13 bullet point 3: This sentence does not make sense, please rephrase.

Page 13, figure7: I don't think this figure is referenced in the text?

Page 14 line 4: "Where the failure region does not encompass the whole ice thickness, an analytical fit was made." This sentence is quite unclear. To my understanding, your results use an analytical fit which is only valid for freeboards less than 1000 m? Is that what was meant here?

Page 14, line 6: The authors conclude that the application to Jakobshavn glacier demonstrates that the modelled calving rate can be "realistic". I'm not sure that the results support a strong conclusion here. The modelled calving rate is lower than the observed calving rate, which is appropriate. But the modelled calving rate could increase by a factor of ten and still meet this condition. I think the discussion needs to be a lot more clear about the very large uncertainties in calving rates produced by this model.

There are also quite a number of spelling and grammar mistakes in the document, and I suggest additional proof reading before resubmission.

Additional references

Bassis, Jeremy N., Sierra V. Petersen, and L. Mac Cathles. "Heinrich events triggered by ocean forcing and modulated by isostatic adjustment." Nature 542.7641 (2017): 332.

Krug, J. W. O. G., et al. "Combining damage and fracture mechanics to model calving." The Cryosphere 8.6 (2014): 2101-2117.

---

## Short Comment (SC1) · 6 Jan 2019

Comment to TCD paper 'A simple stress-based cliff-calving law

Dear authors,
With interest we saw this TCD paper on a cliff-calving-law, in particular as it seems in its approach in many ways very close to our recent co-authored paper in The Cryospere (Mercenier et al 2018, https://doi.org/10.5194/tc-12-721-2018), further referred to as M18.
Both papers use the 2-dimensional full-Stokes stress-field (and scale it) on an idealized rectangular glacier geometry with variable water depth for scaling the location/extent of the stress maxima/critical stress and some failure time related to a damage approach; and both derive a calving 'relation' from that.

Despite its similarity we think this paper may provide a valuable additional contribution to the general development of new calving models as it differs in the details of formulation and expression, criterion and parameters used and in its focus on 'cliff'-calving.

However, we feel this paper lacks a clear and more detailed discussion of the proposed calving-law to other existing approaches and in particular to the similar approach in M18.  It is currently not clear how similar the final calving-relationships and results of the proposed approach really is to M18 or in what way their results differ (the results are unfortunately plotted very differently).
In order to compare the relations I replotted both below in fig. 5 and fig. 7 (the M18 calving rates are from eqn 22 and the failure length from eqn 18 in M18).

These plots clearly quantify how different the two approaches are in their calving rate relationship (see calving rates in fig 7) and also shed some light on why this is so. According to fig 5 it seems to be the approximation of the choice of the failure length (and not the approximation of the failure time) which in this Schlemm-paper is based on the location of the threshold (1Mpa) whereas in M18 it is taken as the location of maximum tensile stress at the surface.
For low freeboards (that are actually observed in nature) this leads to a huge difference (see fig 5 and 7) and the models give only similar results for very high freeboards between 250 and 600m, which we do not think are observed in nature. Question: could one interpret from this that 'cliff'-calving rarely occurs in nature?

This difference for low (realistic) freeboards may also explain the reason why Jakobshavn calving rates are here in this Schlemm-paper strongly underestimated. The proposed calving relation seems to be developed for the 'cliff calving' case, and as stated for relatively high free-boards. But the issue is that such high freeboards (>200m) are rarely observed which questions somewhat the proposed model.

We do not want to claim that the M18 calving model is in anyway more realistic, and we acknowledge that this study (Schlemm et al) has a different focus/purpose, namely on the 'cliff calving'. But we think a more extensive discussion and comparison with M18 (and maybe other models such as Bassis (2011)) would be useful as the approaches are very close in many ways, but different in important points (maybe could also be clarified in the methods p. 7 around line 5).

For the same reason it would also help to better introduce the model of M18 in the introduction, currently it is simply introduced as one calving model within many others with no reference to the similarity in approach to this study.

[Figure]

Fig 7 of Schlemm (TCD) redone: calving rate against freeboard for Schlemm (in review TCD) and M18.

[Figure]

Fig 5 of Schlemm (TCD) redone: showing failure region length (behind front) against freeboard for Schlemm (in review TCD) and M18.

A few other minor editing issues we came across:

- p. 6 line 7: the last sentence missing something (not complete sentence).

- Eqs 15 and 23, it should probably be 0.356 rather than 35.6% to be consistent with figures
- The 'B' from eq 18 (the rate factor for damage evolution) and 21 (effective damage rate) are two different quantities, therefore their notations should be different to avoid confusion (see Eq 22 in M18)

Andreas Vieli, Martin Lüthi and Rémy Mercenier

---

## Author Comment (AC1) · 6 Jan 2019

Dear Andreas, we are terribly sorry. We were not aware of your paper, but we should have. Thank you so much for pointing this out. We will take your comments into account of course. Bests, Anders
* * *

---

## Short Comment (SC2) · 21 Jan 2019

**Authors' reply to Reviewer's comment 1**

**January 21, 2019**

In the paper, Stokes' equation was solved for a flowline model with only one horizontal dimension and with the boundary conditions of a frozen glacier: no inflow at the upstream boundary, u = 0, and no slip at the basal boundary, u = w = 0. The failure region and the cliff calving law were determined for this setup.

The anonymous reviewer pointed out that in most scenarios where cliff calving might happen, the glacier will probably be sliding and experiencing lateral drag, and asked what this means for the derived cliff calving law and whether it could be applied in these cases.

We will investigate these two questions here and find that the derived cliff calving rate can serve as a lower bound for cases with sliding and lateral drag.

**1** Sliding glaciers**

Let's first consider sliding with a constant velocity v (i.e. vanishing strain rate). Then the upstream boundary condition sees an influx with velocity v, so u = v. The basal boundary conditions become u = v, w = 0. Solving the Stokes' equations with these boundary conditions numerically with FeniCS gives the exact same stress fields as in the frozen case (with deviations on the order of magnitude of numerical error, see fig. 1) and the velocity field is simply shifted by the sliding velocity v. This is not surprising: A simple Galilei transformation takes this sliding glacier back to the frozen glacier previously considered without changing any of the physics.

Thus, the assumption made in the paper can be generalized: The cliff calving law derived is valid for glaciers sliding with a constant velocity v with a vanishing strain rate in the last few kilometers before the terminus.

In general, sliding velocities increase towards the glacier terminus. The steepest possible velocity gradient can be obtained with a free-slip basal boundary condition: we assume no influx at the upstream boundary, u = 0, and at the bed we assume free slip in the horizontal direction, which only leaves a boundary condition for the vertical velocity, w = 0. The basal velocity is zero at the upstream boundary and takes its maximum at the calving front. Due to this velocity gradient, the maximum shear stress is large throughout the whole numerical domain (see fig. 2). For increasing ice thickness it becomes difficult to define a meaningful failure region, because the critical shear stress is exceeded in the whole numerical domain one must assume that the whole numerical domain will fail.

Thus, in the case of a non-vanishing velocity gradient, the failure region is larger than in the case of a vanishing velocity gradient. Hence, the derived cliff calving rate can serve as a lower bound for this kind of calving fronts. In case the sliding velocity was decreasing towards the calving front, the failure region and hence the calving rate would be smaller, but that is very

Figure 1: Stresses for the no-slip case, equivalent to sliding with a constant velocity v with vanishing velocity gradient, for three different relative water depths, w = [0, 0.5, 0.85].

unlikely to occur.

To summarize: The derived cliff calving law is valid for glaciers that are frozen to the bed or sliding with a constant velocity and vanishing velocity gradient. It serves as a lower bound on the calving rate for glaciers in which velocities increase towards the calving front.

**2 Lateral drag**

In order to investigate how lateral drag influences cliff calving, we will assume ice flow in a channel with a flowline in the x-direction. Ice is assumed to flow only in the x-direction with a flow maximum in the middle of the channel (see fig. 3). Since deviatoric stresses are connected to the strain rate,  $\tau_{ij} = B\dot{\epsilon}_e\dot{\epsilon}_{ij}$ , and the strain rate is given by the velocity gradients,  $\dot{\epsilon}_{ij} = \frac{1}{2} (\partial_i u_j + \partial_i u_j)$ , we get an additional deviatoric shear stress in the x-y-plane,  $\tau_{xy}$ . The other stress components in y vanish,  $\tau_{yz} = \tau_{yy} = 0$ , because the respective velocity gradients vanish. The Cauchy stress tensor becomes

$$\sigma = \begin{pmatrix} P + \tau_{xx} & \tau_{xy} & \tau_{xz} \\ \tau_{xy} & P & 0 \\ \tau_{xz} & 0 & P - \tau_{xx} \end{pmatrix}$$
(1)

The principal stresses  $\sigma_i$  are defined as eigenvalues of  $\sigma$ , and the maximum shear stress  $\tau_{max}$  as the difference between the maximum and minimum principal stress. In 2D, there is a simple analytical formula for  $\tau_{max}$ . In 3D, there is no simple analytical formula for the eigenvalues of a matrix and therefore it is not feasible to get an analytical estimate on whether

Figure 2: Stresses for the free-slip case for three different relative water depths, w = [0, 0.5, 0.85].

---

## Referee Comment (RC2) · Anonymous Referee #2 · 21 Mar 2019

The objective of the paper is to derive a parameterisation of calving rates for *"cliff-calving"*. *"Cliff-calving"* is defined as calving for ice fronts where the ice thickness exceed a stability limit. However, the paper lack a clear definition for this *"cliff-calving"* making the objective of the paper and applicability of the parameterisation relatively uncertain. The *"cliff-calving"* mechanism has been introduced by Pollard and al. (2015) and De Conto and Pollard (2016) to explain high mass loss rates from the Antarctic in the geological past. However, a recent study (Edwards et al., 2019) shows that this mechanism is not required to reproduce past sea-level changes. These controversial results require improved physically-based models, which is the aim of this paper. As this process is not currently observed, validating these models with observations is, by definition, impossible. It is thus crucial to detail the physical basis of proposed models

and parametrisations and analyse their sensitivity. As pointed by the first reviewer and comments by Vieli and co-authors, the manuscript in his present form is too short on these aspects.

To derive the calving relation, the authors compute the stresses in the vicinity of synthetic ice fronts with various thicknesses and water depth using a full-Stokes ice flow model. A stress criteria (based on the maximal shear stress) is used to define the region that will calve. This is further converted to a calving rate using a reference failure time. As pointed by Vieli and co-authors, this study is extremely similar to Mercenier et al. (2018), and do not really acknowledge it. As Schlemm and Levermann re-use the failure time calibrated by Mercenier et al. (2018), the only difference is the stress criteria and thus the failure region. The first reviewer and Vieli and co-authors, already provide guidance to improve the paper by clarifying the hypotheses, running more sensitivity experiments, comparing with previous similar studies and improving the discussion to define the applicability of the proposed calving law. I fully support their main comments and this implies major changes in paper.

Finally, at the end, Jackobsahvn is presented as one of the few glaciers that is *"in the calving cliff regime"*; However, this *"cliff regime"* is not really defined, from page 7 lines 29-30, I understand that the authors define the cliff regime from their critical shear stress; So a glacier would be in the cliff regime if their critical shear stress is reached somewhere in the domain; which from their numerical experiments appends only for freeboards larger than 100m? At the end it is a bit disappointing that the proposed parameterisation underestimate the calving rate of one the few glaciers in what the authors call the *"calving cliff regime"*, by more than one order of magnitude. Especially when the parametrisation from Mercenier et al. (2018) does a fairly good job for the same glacier. As shown by Vieli, the Schlemm and Levermann caving rates become higher than the Mercenier et al. calving rates for larger free boards. So the paper should really focus on giving better description and justification for their mechanism, and its domain of applicability. Should it replace existing parameterisations for large

freeboards? In this case how to define the transition to the cliff regime? Should we sum the processes or take the maximum calving rate? Without answering theses question properly I don't see how the proposed parametrisation could be used by the community.

I give few more detailed comments on the paper below :

- Abstract: the mechanism "cliff-calving" is not really defined in the abstract and there is a confusion with "normal" calving of tide water glaciers as currently observed, see comment above. This distinction and the definition of "cliff calving" is also not really clear in the introduction. It should be clear since the beginning that the paper propose an extension (extrapolation) of the calving mechanism to glacier freeboard heights that are not currently observed.

- Page 1, lines 18-22: the word "loss" introduces a confusion between the processes that remove ice from the ice sheets (what is implied with the reference to Antarctica), and the fact that the ice sheets are not in balance due to increase losses by calving and/or melt (the numbers for Greenland are the respective contribution to the unbalance). Please clarify.

- "Failure region" everywhere in the text and Figures 3-4-5. There is a confusion between the region where the stress is higher than the threshold and the "failure distance" L. It seems that L is the maximum distance from the front where the stress is in excess to the critical stress. Please clarify.

- Figure 4. What is the color scale? Indicate that the outline for H=1000m is also shown in Fig. 3 (top-left).

- Page 9, line 4: clarify the "bend" and the "two fits" at the critical freeboard.

- Page 9, Eqs. 14-17: explain the values for the fit; which ones have been optimised, which ones are prescribed and why?

- Page 11: comparison with Jackobsahvn; clarify the discussion about the grounding line and front.

---

## Author Response (AR1)

**Authors' response**

We thank the referees for their comments and questions that have helped us improve the manuscript.

We have:

- clarified the Jakobshavn example,
- compared our results with Mercenier et.al (2018) in the introduction as well as the discussion,
- discussed under what conditions the derived cliff calving relation is valid (sliding, lateral drag) and
- extended the discussion of time to failure with a focus on uncertainties.

Following, we give detailed replies to the referees' comments.

**Anonymous Referee #1**

**Major comments**

The model used to derive the proposed analytical calving law has a highly simplified geometry – with zero surface or bed slope, no lateral drag, and no sliding at the bed. Previous studies have shown that the extent of the "failure region" discussed in this paper is strongly affected by basal sliding rates (Ma et al., 2017). Likewise, other studies have shown that the stress regime around the calving front is strongly affected by surface slope (Mercenier

studies have shown that the stress regime around the calving front is strongly affected by surface slope (Mercenier et al, 2018).

As the authors point out, "there are no glaciers currently available where cliff calving is the primary failure mechanism", but modelling studies such as DeConto and Pollard (2016) suggest cliff failure could occur in future in deep Antarctic basins, after rapidretreat of their buttressing ice shelves. These environments are highly likely to experience basal sliding, as well as lateral drag. It is hard to say what proportion of ice cliffs might meet the authors' conditions, but the proposed model for predicting calving rates seems a lot less generally applicable than simply using the maximum shear stress to define a new calving front location. At the very least the paper should include

more discussion of precisely what circumstances the model is valid for, and under what conditions (e.g. basal sliding) it is likely to fail.

**Response:** We thank the reviewer for this valuable suggestion. We gave a lengthy reply to the effect of sliding and lateral drag in the short comment answering this review and have included a discussion in the manuscript (section 7.1)

**Minor comments**

*Page 2 line 6: Columbia glacier is in Alaska, not Canada* **Response:** Thank you, this has been corrected.

Page 2 lines 28-29: The description of Mercenier et al. (2018) is extremely brief and doesn't contrast the model with other studies, which would be much more informative. This also seems a suitable point to reference Morlighem et al. (2016).

**Response:** Done (p.2, l.27; p.2, l.33 – p.3., l.2). A more in depth discussion of Mercenier is given in the discussion (section 7.3.2)

*Introduction: The introduction misses damage mechanics methods which have been used to implement calving in a tidewater glacier (Krug et al., 2014).* **Response:** Done (p.2, l.25-27).

*Page 3 line 5: "it is not clear what a cliff calving law would look like". Are the authors aware of Bassis et al. (2017) which already implemented a calving law based on cliff instability?*

**Response:** Thank you for pointing this out, the reference has been included (p.3, l.5).

*Page 4 lines 5-10: No boundary condition is provided for the upstream boundary of the model (r.h.s. in figure 1)* **Response:** The upstream boundary condition has been added. No inflow is assumed at this boundary.

*Page* 5, *eq.* 11: *Should y in this equation be z*? **Response:** Yes, it has been corrected.

Page 7 lines 2-3: "However, it does not take into account whether deviatoric stresses are tensile or compressive or shear stresses and this is likely to be important for ice failure." Surely this is the advantage of using the von Mises stress as a criterion – it is able to allow for failure under both tension and shear, and is therefore more widely applicable than a criterion that considers only one mechanism of failure?

**Response:** The reviewer makes a valuable point. However, the maximum shear stress and the von Mises stress differ only by a factor of  $\sqrt{3}$ . Choosing the von Mises stress instead of the maximum shear stress as the failure criterion would not change the results qualitativley. We chose the shear stress because it gives a more clear physical explanation of how the failure happens. This discussion has been added to the manuscript (p.6, 1.9-10; 1.25-26).

*Page 7 lines 5-10: these uncertainties should be explored further in the discussion, which doesn't currently make their magnitude clear.*

**Response:** Estimate of the magnitude has been given (p.7, l.1 and section 7.2).

*Page 7 eq. 13: I'm not sure where the term sqrt*( $\mu^2$ +1) *on the l.h.s. comes from here.*

**Response:** Me neither, the derivation is not given in the cited literature. Eq. 12 expresses the failure condition in terms of the stresses along the future fault plane and therefore depends on the direction of the fault plane. Eq. 13 gives a more general expression of the failure condition in terms of the maximum shear stress and the isotropic pressure. This has been clarified in the manuscript (p.8, l.1).

*Page* 9 *line* 1: "Above a critical freeboard of about 1000m the failure region encompasses the whole ice thickness." Is this based on results from figure 4? **Response:** Yes. This has been added in the manuscript (p.8, 1.23).

Page 9 lines 3-4: "The freeobard [sic]- failure region relation has a bend at the critical freeboard and hence the two parts require separate analytical fits" Figure 5 shows no freeboards above 800 m, so readers cannot see how this conclusion was reached.

**Response:** The figure has been modified to show larger freeboards.

Page 10 eq. 18: What are k, r and B?

**Response:** k, r and B are material constants. This has been added in the manuscript (p.10, l.9).

*Page 11 eq. 19 & 20: I think*  $\sigma$  *and*  $\sigma$ *0 here are not the same as in previous equations?* **Response:** No, they are not.  $\sigma$  is the major stress and  $\sigma$ 0 is the instantaneous strength. This has been added in the manuscript (p.12, l.5).

*Page 11 eq. 21: Is part of this equation missing? What values have you used for k, D0 and Dc?* **Response:** All the material properties have been included in B, which has been renamed B\* to avoid confusion.

*Page 13 bullet point 3: This sentence does not make sense, please rephrase.* **Response:** Done.

*Page 13, figure7: I don't think this figure is referenced in the text?* **Response:** Done.

Page 14 line 4: "Where the failure region does not encompass the whole ice thickness, an analytical fit was made." This sentence is quite unclear. To my understanding, your results use an analytical fit which is only valid for freeboards less than 1000 m? Is that what was meant here? **Response:** Yes, this was clarified (p.15, l.7).

Page 14, line 6: The authors conclude that the application to Jakobshavn glacier demonstrates that the modelled calving rate can be "realistic". I'm not sure that theresults support a strong conclusion here. The modelled calving rate is lower than the observed calving rate, which is appropriate. But the modelled calving rate could increase by a factor of ten and still meet this condition. I think the discussion needs to be a lot more clear about the very large uncertainties in calving rates produced by this model.

**Response:** We use Jakobshavn to show that cliff calving rates are not overestimated (p.13, l.3-5). A discussion about the uncertainties has been included (section 7.2).

There are also quite a number of spelling and grammar mistakes in the document, and I suggest additional proof reading before resubmission.

**Response:** Spelling and grammar have been checked.

**Anonymous Referee #2**

**Major comments**

To derive the calving relation, the authors compute the stresses in the vicinity of synthetic ice fronts with various thicknesses and water depth using a full-Stokes ice flow model. A stress criteria (based on the maximal shear stress) is used to define the region that will calve. This is further converted to a calving rate using a reference failure time. As pointed by Vieli and co-authors, this study is extremely similar to Mercenier et al. (2018), and do not really acknowledge it. As Schlemm and Levermann re-use the failure time calibrated by Mercenier et al. (2018), the only difference is the stress criteria and thus the failure region. The first reviewer and Vieli and co-authors, already provide guidance to improve the paper by clarifying the hypotheses, running more sensitivity experiments, comparing with previous similar studies and improving the discussion to define the applicability of the proposed calving law. I fully support their main comments and this implies major changes in paper.

**Response:** The section about the time to failure has been expanded to clarify that the time to failure Mercenier et al. (2018) derived for tensile failure might not be suitable to shear failure, but is used nevertheless because there are no better guesses available. The discussion section has been expanded significantly to include discusion about the effect of sliding and lateral drag on the cliff calving rate (section 7.1), uncertainties in the time to failure (section 7.2), as well as a comparison with other cliff calving parametrisations and with the study of Mercenier et al. (2018) (section 7.3).

Finally, at the end, Jackobsahvn is presented as one of the few glaciers that is "in the calving cliff regime"; However, this "cliff regime" is not really defined, from page 7 lines 29-30, I understand that the authors define the cliff regime from their critical shear stress; So a glacier would be in the cliff regime if their critical shear stress is reached somewhere in the domain; which from their numerical experiments appends only for freeboards larger than 100m? At the end it is a bit disappointing that the proposed parameterisation underestimate the calving rate of one the few glaciers in what the authors call the "calving cliff regime", by more than one order of magnitude. Especially when the parametrisation from Mercenier et al. (2018) does a fairly good job for the same glacier.

**Response:** A glacier enters the cliff calving regime, when its freeboard is larger than the critical freeboard Fc and the cliff calving rate given by eq. 22 becomes nonzero. This definition has been added to the text (p.13, l.3) and the discussion has been extended to clarify that we expect cliff calving (i.e. shear failure) to play a role in Jakobshavn but since the freeboard is still rather small and the glacier is heavly crevassed, we expect tensile failure to be the main contribution to the overall calving rate. The Jakobshavn example is meant to be a "sanity check" and can only give an upper bound on the cliff calving rate.

As shown by Vieli, the Schlemm and Levermann caving rates become higher than the Mercenier et al. calving rates for larger free boards. So the paper should really focus on giving better description and justification for their mechanism, and its domain of applicability. Should it replace existing parameterisations for large freeboards? In this case how to define the transition to the cliff regime? Should we sum the processes or take the maximum calving rate? Without answering theses question properly I don't see how the proposed parametrisation could be used by the community.

**Response:** Unfortunately, we cannot give a definite answer to these questions. We expect tensile failure to dominate for small freeboards and shear failure to dominate for large freeboards. However, it is difficult to say at which glacier freeboard the tensile failure regime ends and the shear failure regime begins, not only due to uncertainty in the scaling parameter C0. In practice, both failure modes will interact, with tensile stress damaging the ice through few large crevasses originating from the surface of the ice and shear stress damaging the ice through a large number of small fractures in the lower part of the cliff. This likely interaction of failure modes cannot be analyzed by assuming ice to be a continuous medium (like the approach used here and by Mercenier et al. (2018)), but should be done with damage theory or a discrete element approach. This is discussed in section 7.3

**Minor comments**

Abstract: the mechanism "cliff-calving" is not really defined in the abstract and there is a confusion with "normal" calving of tide water glaciers as currently observed, see comment above. This distinction and the definition of "cliff calving" is also not really clear in the introduction. It should be clear since the beginning that the paper propose an extension (extrapolation) of the calving mechanism to glacier freeboard heights that are not currently observed.

**Response:** This was clarified in the abstract.

Page 1, lines 18-22: the word "loss" introduces a confusion between the processes that remove ice from the ice sheets (what is implied with the reference to Antarctica), and the fact that the ice sheets are not in balance due to increase losses by calving and/or melt (the numbers for Greenland are the respective contribution to the unbalance). Please clarify.

**Response:** This has been clarified.

"Failure region" everywhere in the text and Figures 3-4-5. There is a confusion between the region where the stress is higher than the threshold and the "failure distance" L. It seems that L is the maximum distance from the front where the stress is in excess to the critical stress. Please clarify. **Response:** This was clarified (section 4)

*Figure 4. What is the color scale? Indicate that the outline for H*=1000*m is also shown in Fig. 3 (top-left).* **Response:** An explanation of the color scale and mention of the outline has been added to the figure caption.

*Page 9, line 4: clarify the "bend" and the "two fits" at the critical freeboard.* **Response:** Done (p.8, 1.25-27).

Page 9, Eqs. 14-17: explain the values for the fit; which ones have been optimised, which ones are prescribed and why?

**Response:** At first L was fitted as a function of F for each value of w. Then the parameter functions Fs,Fc and s were fitted as functions of w. This has been added to the manuscript (p.10, l.1-2).

*Page 11: comparison with Jackobsahvn; clarify the discussion about the grounding line and front.* **Response:** Done (p.13, l. 6-10).

**A simple stress-based cliff-calving law**

Tanja Schlemm1,2 and Anders Levermann1,2,3

1Potsdam Institute for Climate Impact Research, Potsdam, Germany
 2Institute of Physics and Astronomy, University of Potsdam, Potsdam, Germany
 3Lamont-Doherty Earth Observatory, Columbia University, New York, USA

Correspondence: Anders Levermann (anders.levermann@pik-potsdam.de)

Abstract. Over large coastal regions in Greenland and Antarctica the ice sheet calves directly into the ocean. In contrast to ice-shelf calving, an increase in [..1] calving from grounded glaciers contributes directly to sea-level rise[..2]. Ice cliffs with a glacier freeboard larger than  $\approx 100$  m are currently not observed, but it has been shown that such ice cliffs are increasingly unstable with increasing ice thickness. This cliff calving can constitute a self-amplifying ice loss mechanism

5 that may significantly alter sea-level projections both of Greenland and Antarctica. Here we seek to derive a [..3]minimalist stress-based [..4]parametrization for cliff calving [..5]from grounded glaciers whose freeboards exceed the 100 m stability limit derived in previous studies. This will be an extension of existing calving laws for tidewater glaciers to higher ice cliffs.

To this end we compute the stress field for a glacier with a simplified two-dimensional geometry from the two-dimensional

- 10 Stokes equation. First we assume a constant yield stress to derive the failure region at the glacier front from the stress field within the [..6]glacier. Secondly, we assume a constant response time of ice failure due to exceedance of the yield stress. With this strongly constraining but very simple set of assumption we propose a cliff-calving law where the calving rate follows a power-law dependence on the freeboard of the ice with exponents between 2 and 3 depending on the relative water depth at the calving front. The critical freeboard below which the ice front is stable decreases with increasing relative water depth of the
- 15 calving front. For a dry water front it is, for example,  $[..^7]75$  m. The purpose of this study is not to provide a comprehensive calving law, but to derive a particularly simple equation with a transparent and  $[..^8]$  minimalist set of assumptions.

Copyright statement. ...

- 7removed: 75m
- 8removed: minimalistic

<sup>1removed: cliff calving directly contributes

<sup>2removed: and a monotonously increasing calving rate with ice thickeness

<sup>3removed: minimalistic

<sup>4removed: parameterization

<sup>5removed: .

<sup>6removed: ice sheet

**1 Introduction**

[revised manuscript text omitted]

---

## Author Response (AR2)

[revised manuscript text omitted]

**References**

[revised manuscript text omitted]